

# Chemical ionisation quadrupole mass spectrometer with an electrical discharge ion source for atmospheric trace gas measurement

Philipp G. Eger[1], Frank Helleis[1], Gerhard Schuster[1], Gavin J. Phillips[1,2], Jos Lelieveld[1] and John N. Crowley[1]

[1]Atmospheric Chemistry Department, Max-Planck-Institut für Chemie, 55128 Mainz, Germany
[2]Department of Natural Sciences, University of Chester, CH2 4NU, UK

*Correspondence to*: John N. Crowley (john.crowley@mpic.de)

**Abstract.** We present a Chemical Ionisation Quadrupole Mass Spectrometer (CI-QMS) with radio-frequency (RF) discharge ion source through $N_2$ / $CH_3I$ as source of primary ions. In addition to the expected detection of PAN, peracetic acid and $ClNO_2$ through well-established ion-molecule-reactions with $I^-$ and its water cluster, the instrument is also sensitive to $SO_2$, HCl and acetic acid ($CH_3C(O)OH$) through additional ion chemistry unique for our ion source. We present ionisation schemes for detection of $SO_2$, HCl and acetic acid along with illustrative data sets from three different field campaigns underlining the potential of the CI-QMS with an RF discharge ion source as an alternative to $^{210}Po$. The additional sensitivity to $SO_2$ and HCl makes the CI-QMS suitable for investigating the role of sulphur and chlorine chemistry in the polluted marine and coastal boundary layer.

## 1 Introduction

Chemical Ionisation Mass Spectrometry using iodide anions (commonly referred to as I-CIMS) is a widely used technique to measure various atmospheric trace gases with high temporal resolution and low detection limits. The potential of I-CIMS for atmospheric trace gas measurement was established in laboratory studies (Huey et al., 1995) on chlorine nitrate ($ClONO_2$) which plays a central role in polar stratospheric $O_3$ depletion (Molina et al., 1987) and dinitrogen pentoxide ($N_2O_5$) which, through its heterogeneous hydrolysis on cloud droplets and aerosols, acts as a sink of gas-phase $NO_x$ ($NO + NO_2$) (Lelieveld and Crutzen, 1990; Dentener and Crutzen, 1993). The first applications of I-CIMS for monitoring atmospheric composition were for measurement of $N_2O_5$, peroxyacyl nitric anhydride (PAN, $CH_3C(O)O_2NO_2$) and other peroxycarboxylic nitric anhydrides (Huey, 2007). Since then, it has been recognised that several classes of organic and inorganic traces gases can be sensitively detected by I-CIMS including organic and inorganic acids, organic nitrates, halogen-nitrates and di-halogens (Phillips et al., 2013; Lee et al., 2014; Neuman et al., 2016; Priestley et al., 2018). Our instrument has previously been deployed with a radioactive ion source ($^{210}Po$) to investigate the atmospheric chemistry of nitryl chloride ($ClNO_2$), PAN and peracetic acid (PAA, $CH_3C(O)OOH$) (Phillips et al., 2012; Phillips et al., 2013; Phillips et al., 2016; Crowley et al., 2018).



PAN (and other peroxycarboxylic nitric anhydrides) are formed in the atmosphere via the reaction of $NO_2$ with peroxyacyl radicals ($RC(O)O_2$) generated during the photo-oxidation of VOCs. Slusher et al. (2004) reported the first detection of PAN, PPN (peroxypropionic nitric anhydride), MPAN (peroxymethacrylic nitric anhydride) and PiBN (peroxyisobutyric nitric anhydride) using thermal decomposition chemical ionisation mass spectrometry (TD-CIMS) with iodide ions. The most abundant, PAN, is of great importance owing to its role in transportation of $NO_2$ from source regions to remote areas (Moxim et al., 1996). The detection of PAN and its analogues via I-CIMS requires thermal dissociation (generally at temperatures close to 100 °C) to the peroxy radical, which then reacts with $I^-$ primary ions to form the carboxylate anion, which is detected. Compared to gas chromatographic methods for detection of peroxycarboxylic nitric anhydrides the I-CIMS technique allows faster measurements with comparable sensitivity and selectivity (Slusher et al., 2004; Roiger et al., 2011) enabling eddy covariance flux measurements (Turnipseed et al., 2006; Wolfe et al., 2009). Although the bond dissociation energy of the peroxycarboxylic nitric anhydrides are similar, Zheng et al. (2011) report lower sensitivity for APAN (peroxyacrylic nitric anhydride), PiBN, PnBN (peroxy-n-butyric nitric anhydride) and CPAN (peroxycrotonyl nitric anhydride) and Mielke and Osthoff (2012) report lower sensitivity for MPAN compared to PPN and PAN. Peroxycarboxylic nitric anhydrides have been measured using I-CIMS in various locations including the boreal forest (Phillips et al., 2013), pine forests (Turnipseed et al., 2006; Wolfe et al., 2009), urban areas (Slusher et al., 2004; LaFranchi et al., 2009; Wang et al., 2017) and the Arctic (Roiger et al., 2011).

Detection of PAA by I-CIMS was reported by Phillips et al. (2013) who performed the first combined measurement of PAN and PAA in the boreal forest in Finland. PAA acts as a significant sink for $CH_3C(O)O_2$ and $HO_2$ under low-$NO_x$ conditions and can compete with PAN formation especially at high temperatures (Crowley et al., 2018). As for PAN, PAA was detected as the acetate anion at a mass-to-charge ratio (*m/z*) of 59. At this *m/z*, the I-CIMS deployed by Phillips et al. (2013) was insensitive to acetic acid. A wide range of organic acids can be detected as an $I^-$ cluster with the parent molecule (Le Breton et al., 2012; Lee et al., 2014) using time of flight mass spectrometers (I-CIMS-TOF) which have high mass resolution and which exploit the iodine mass defect for identification of the elemental composition of the organic trace gases detected.

$ClNO_2$ is formed in the heterogeneous reaction of $N_2O_5$ on chloride containing particles and surfaces during night (Behnke et al., 1997). The day-time photolysis of $ClNO_2$ results in the release of chlorine atoms, which enhance oxidation rates of organic trace gases especially during early morning hours (Phillips et al., 2012; Riedel et al., 2012). Nitryl chloride has been observed by I-CIMS as $IClNO_2^-$ and $ICl^-$ in the polluted marine boundary layer (Osthoff et al., 2008; Riedel et al., 2012), close to the coast, e.g. in Hong-Kong (Wang et al., 2016) and London (Bannan et al., 2015) but also inland in continental Northern America (Thornton et al., 2010; Mielke et al., 2011; Faxon et al., 2015), rural continental Europe (Phillips et al., 2012) and Northern China (Tham et al., 2016).

Most I-CIMS systems in operation for atmospheric measurement use a radioactive ion source (usually [210]Polonium, an α-emitter) to generate the primary iodide ions from methyl iodide ($CH_3I$). Although this type of ion source is well-established and known for its high stability and low chemical background, an important, and sometimes unsurmountable obstacle to its use are safety regulations for shipment, storage and operation of radioactive devices containing Polonium. Potential



alternatives are corona discharge and x-ray ion sources as commonly used in atmospheric pressure chemical ionisation mass spectrometers (AP-CIMS) (Jost et al., 2003; Skalny et al., 2007; Kürten et al., 2011; Wang et al., 2017). However, discharge or x-ray ion sources for generating iodide ions at reduced pressure are rarely reported in literature as they often suffer from high chemical background.

We have developed an CI-QMS instrument (Chemical Ionisation Quadrupole Mass Spectrometer) with an electrical discharge ion source that generates iodide ions without use of a radioactive ioniser. Although this instrument was originally intended for measurement of PAN, PAA and $ClNO_2$, we discovered that a wider variety of gas-phase species, including $SO_2$, HCl and acetic acid could be detected. In the following we show that the instrument is suitable for measurement (at the tens of pptv level) of trace gases connected with sulphur and chlorine chemistry, e.g. in the anthropogenically influenced marine

boundary layer. Its deployment as a PAN detector is limited to environments where PAN mixing ratios regularly exceed hundred pptv or when high temporal resolution is not necessary.

## 2 Instrumentation

Our Chemical Ionisation Quadrupole Mass Spectrometer (CI-QMS) is based on the thermal dissociation technique described by Slusher et al. (2004) and Zheng et al. (2011) and was originally constructed in collaboration with Georgia Tech as a

prototype THS Instrument (http://www.thsinstruments.com). A schematic diagram of the instrument in its present form is given in Fig. 1. The major modification, forced by issues of restricted use of polonium on some platforms, is replacement of the $^{210}$Po-ioniser with an electrical discharge ion source (see Fig. 2). The instrument as sketched in Fig. 1 consists of a Thermal Dissociation region (TDR), Discharge Ion Source (DIS), Ion Molecule Reactor (IMR), Collisional Dissociation Chamber (CDC), Octopole ion guide (OCT), Quadrupole Mass Filter (QMF) and Detector (DET). The four different vacuum

chambers are separated by critical orifices and pumped by a combination of scroll pumps and turbo-molecular pumps. The CI-QMS is built into an aircraft rack (HALO, Gulfstream G 550) to facilitate airborne operation. A description of the various parts of the instrument as identified in Fig. 1 is given in the following.

### 2.1 Thermal Dissociation Region (TDR)

During standard operation a flow ($F_1$) of 2.2 L (STD) min$^{-1}$ (slm) is sampled through the instrument inlet (see Fig. 1). The

thermal decomposition of PAN takes place in a 20 cm length of PFA tubing (9.3 mm ID) enclosed in a snug-fitting stainless steel shell heated to 200 °C. A gas temperature of 160–170 °C was measured inside the heated section of tubing. The last 4 cm long section of PFA in front of the orifice to the IMR is not actively heated but the gas temperature still remains at ≈ 130 °C, suppressing PAN recombination. Due to space restrictions inside the aircraft rack, the oven is curved over a 90° bend, however no significant reduction in sensitivity due to a loss of $CH_3C(O)O_2$ on the PFA walls could be observed when

compared to a straight TD-inlet. A bypass flow of 1 slm ($F_2$) decreases the inlet residence time and therefore optimises the peroxyacyl radical transmission. However, as PAN is calibrated in-situ with a photochemical source (see Sect. 2.7) the fractional transmission of $CH_3C(O)O_2$ does not need to be known. 1.2 slm ($F_1$ minus $F_2$) of the inlet flow enter the IMR via a



constant-pressure orifice (see Sect. 2.3) and mix with the 0.8 slm flow ($F_3$) of $CH_3I$ / $N_2$ passing through the ion source. For ground-level deployment, the TDR is held at ambient pressure resulting in a residence time of $\approx$ 200 ms at 1 bar. The rate coefficient for the thermal decomposition of PAN (at 453 K and 1 bar) is $\approx$ 2000 s$^{-1}$ (Atkinson et al., 2006; IUPAC, 2018), so that > 99.99 % of PAN should be thermally dissociated within 200 ms. This could be confirmed by measurement of the

signal due to a stable PAN source whilst varying the inlet temperature. For application in an aircraft the arrangement of TDR and constant-pressure orifice can be switched so that a constant pressure of $p_1$ = 100 hPa is established in the TDR resulting in 40 ms residence time. With this mode of operation measurement can be made at altitudes up to $\approx$ 15 km depending on aircraft inlet configuration. In order to discriminate between PAN and PAA (both measured at *m/z* 59) in ambient air a known concentration of NO is added at the front end of the TDR to remove the $CH_3C(O)O_2$ radicals formed after thermal

decomposition of PAN (see Sect. 4.1).

## 2.2 Discharge Ion Source (DIS)

The radio-frequency (RF) discharge ion source represents the major difference to I-CIMS instruments commonly described in the literature. It consists of two tungsten needles, their tips placed at a distance of approx. 6 mm (adjustable) from each other (see Fig. 2); changing this distance by a few millimetres did not have a large effect on the overall ion count rate. The

vacuum fittings through which the needles enter the discharge volume can be optionally evacuated to eliminate an air-leak into the ion source. A 2.5 kV voltage (20 kHz) applied across the tungsten needles leads to the formation of a stable glow discharge (3 mA). An RF discharge was chosen for its advantages for operation in negative ion mode and because it is easier to handle with regard to electric field geometry and polarity. The discharge between the tungsten needles can be observed by eye or spectroscopically through a quartz viewing port; a photograph of the glow and the dispersed $N_2$ emission spectrum

due to the $B^3\Pi_g \leftarrow C^3\Pi_u$ transition (Lofthus and Krupenie, 1977; Bayram and Freamat, 2012) is shown in Fig. S1 of the supplementary information. The line intensities increase with the voltage applied ($\approx$ factor 2.5 from 1500 to 3000 V) but the relative intensities do not change significantly. From the relative line intensities we calculate that the $N_2$ molecules have a vibrational temperature of $\approx$ 3000 K (Wang et al., 2017).

In normal operation, a flow of 0.8 slm ($F_3$) of 2 ppmv methyl iodide ($CH_3I$) in nitrogen passes through the ionisation region.

In order to prevent the back-flow of air from the IMR into the discharge region, the flow through the ionisation region is kept high (0.8 slm) and passes through a 0.9 mm aperture before entering the IMR.

Similar to the α-radiation of a $^{210}$Po-ioniser, I$^-$ ions are formed via dissociative electron attachment to $CH_3I$. In contrast to typical corona ion sources with a current of a few μA (e.g. see Kürten et al. (2011)) our glow discharge operates at $\approx$ 3 mA, which leads to highly-energetic electrons and ions in the ionisation region and a more complex mass spectrum (see Sect. 3).



## 2.3 Ion Molecule Reactor (IMR)

Under standard operating conditions for ground-level measurements, a flow of 1.2 slm of the air to be analysed ($F_1$ minus $F_2$) is mixed with the 0.8 slm flow of $CH_3I$ / $N_2$ ($F_3$) entering the IMR from the ion source. The IMR is evacuated by a dry scroll vacuum pump (ULVAC DISL-101, 100 l min$^{-1}$) and is held at a constant pressure of 24 mbar ($p_2$). For operation above the

boundary layer, an extra 50 cm$^3$ (STD) min$^{-1}$ (sccm) flow of humidified air is added just in front of the IMR to ensure that sufficient water vapour is present to form $I^-(H_2O)$ clusters. The role of $I^-(H_2O)$ and other primary ion clusters with water is discussed in Sect. 5.

The constant pressure within the IMR is achieved by use of a variable orifice consisting of two metal plates, one with a hole shaped like the 2D-projection of a bike saddle and one with circular hole, the relative position of which (i.e. the degree of

overlap of the holes) is controlled by a stepper motor. The saddle form was chosen as it results in a roughly linear relationship between stepper motor position and mass flow through the orifice, enabling rapid and precise adaptation to changes in ambient pressure even during steep ascents or dives of an aircraft.

The temperature in the IMR is above ambient owing to the inflow of heated gas through the TDR. The exact residence time for trace gases to react with primary ions in the IMR is not accurately known as the mixing of the gas flows and temperature

evolution in the IMR are not well characterised. Based on the mass flow rate into the IMR and its volume ($\approx$ 100 cm$^3$) and disregarding ion-drift due to the potential applied between the IMR and the CDC we calculate an approximate reaction time of $\approx$ 70 ms.

## 2.4 Collisional Dissociation Chamber (CDC) and Octopole ion guide (OCT)

The CDC region consists of an octopole ion guide to accelerate and collimate the effusive ion beam entering from the IMR.

It is separated from the IMR by a critical orifice (0.8 mm diameter) and held at a pressure of 0.6 mbar ($p_3$), by the Holweck stage of a turbo-molecular pump (Leybold Turbovac 90i, 90 l s$^{-1}$ with Agilent IDP-3 scroll pump, 50 l min$^{-1}$ as back-up pump). A potential difference of typically 20 V is applied in the CDC, which results in the de-clustering of weakly-bound adducts (often with $H_2O$) and results in a simplified mass spectrum and a higher sensitivity for the product ion of interest. The de-clustering voltage can be varied independently for each ion of interest, thus optimising sensitivity for individual trace

gases. An example of the variable de-clustering potentials whilst operation of the CI-QMS in selected ion monitoring mode (e.g. to measure the $I^-(H_2O)$ cluster or differentiate between acetic and peracetic acid) is given later.

An additional octopole ion guide in the subsequent vacuum chamber ($6.0 \times 10^{-3}$ mbar, $p_4$) further collimates the ion-beam and guides it to the detector region. This octopole ion guide is evacuated by the turbo-molecular stage of the Leybold Turbovac 90i.





## 2.5 Quadrupole Mass Filter (QMF) and Detector (DET)

A radio-frequency generator (Balzers QMH 410-3, 1.44 MHz) provides a combination of direct and alternating voltage to the quadrupole rods (10 mm) so that ions with a specific *m/z* are forced on stable trajectories and reach the detector. These ions are then detected with a channel electron multiplier (ITT Ceramax 7550M). The detector chamber is pumped to a

pressure of $9.0 \times 10^{-5}$ mbar ($p_5$) by a turbo-molecular pump (Varian V70LP, $70\,l\,s^{-1}$).

## 2.6 Scrubber

To determine the instrumental and chemical background the sampled air is automatically and periodically bypassed into a scrubber (see Fig. 1) consisting of a 20 cm long stainless steel oven filled with steel wool and heated to 120 °C. The trace gases discussed in this work are all destroyed efficiently by the hot metal surfaces whilst leaving the relative humidity

unaffected.

## 2.7 Photochemical PAN source

For in-situ PAN calibration we use a photochemical source based on the method of Warneck and Zerbach (1992) but with a phosphor-coated Pen-Ray mercury lamp (Jelight, broad emission centred at 285 nm) as described by Flocke et al. (2005). Typically, 50 sccm of acetone (200 ppmv in synthetic air, Air Liquide) and 5 sccm of NO (1 ppmv in $N_2$, Air Liquide) are

mixed in a quartz glass reactor (150 ml volume, actively cooled by a fan) at 1050 mbar forming PAN (Reactions R1–R4). The calibration source converts NO almost stoichiometrically to PAN (conversion > 95 %) and results in a mixing ratio of about 4 ppbv of PAN in the TDR. The PAN source is continuously operated and its 55 sccm output drains into the exhaust line, with periodically switching into the main flow during scrubbing. The conversion efficiency of NO to PAN was checked using thermal dissociation cavity ring-down spectroscopy as described previously (Phillips et al., 2013). The PAN source

also generates both peracetic acid (PAA) and acetic acid (R5–R6), which, as described later, are also detected by the CI-QMS.

| | | | |
|---|---|---|---|
| $CH_3C(O)CH_3 + h\nu\ (O_2)$ | $\rightarrow$ | $CH_3O_2 + CH_3C(O)O_2$ | (R1) |
| $CH_3O_2 + NO\ (O_2)$ | $\rightarrow$ | $HCHO + HO_2 + NO_2$ | (R2) |
| $CH_3C(O)\,O_2 + NO\ (O_2)$ | $\rightarrow$ | $CH_3O_2 + CO_2 + NO_2$ | (R3) |
| $CH_3C(O)O_2 + NO_2 + M$ | $\rightarrow$ | $CH_3C(O)O_2NO_2 + M$ | (R4) |
| $CH_3C(O)O_2 + HO_2$ | $\rightarrow$ | $CH_3C(O)OOH + O_2$ | (R5) |
| | $\rightarrow$ | $CH_3C(O)OH + O_3$ | (R6) |



## 2.8 Electronics and data acquisition

The vast majority of the instrument's electronics is controlled by a "V25" system developed in-house. The V25 handles the interplay between single components such as flow controllers, pressure gauges, magnetic valves, thermocouples, heaters, MS potentials and RF generator. All command sequences and measurement cycles (background, calibration etc.) can be customised and fully automated for operation in aircraft or in remote locations. During measurement campaigns we usually focus on specific trace gases and operate in selected ion monitoring mode, typically monitoring between 3 and 10 $m/z$ to increase the temporal resolution. The ion movement inside the quadrupole is determined by Mathieu differential equations and different $m/z$ values can be adjusted in a few ms by variation of the direct voltage and the amplitude of the alternating voltage applied to the quadrupole rods. The integration time of the detector for a single channel is usually set to 10 ms for primary ions and 100 ms for product ions, which represents a compromise between high signal-to-noise ratio (S/N) and high temporal resolution. For each $m/z$ monitored, the integrated signal is calculated by summing up 8 individual channels, resulting in a temporal resolution for one single molecule of about 800 ms. Higher frequency measurements are possible at the cost of a reduction in the signal-to-noise ratio. The counts for each channel and the integrated counts as well as the most important system parameters are saved on an internal PC card and can be additionally collected and monitored online using customised LabView software. To identify additional traces gases of interest, the whole mass spectrum ($m/z$ 1–256) is occasionally scanned and recorded, which takes about 1–2 minutes.

## 2.9 Size, weight and power consumption

The CI-QMS is situated in a compact aircraft rack (65 x 55 x 140 cm, HALO, Gulfstream G 550) with a total weight of 135 kg and a power consumption of 0.9 kW with the vacuum pumps as main power consumers. The two vacuum scroll pumps require 230 V AC input whereas all the other components are operated with 24 V DC from an AC/DC converter that either can be supplied with 230 V AC or 115 V AC (3 phase, 400 Hz, for aircraft operation).

## 3 Primary-ion spectra

The deployment of an RF discharge source for iodide-ion production leads to a more complex primary-ion mass spectrum when compared to use of $^{210}$Po; consequently, a wider variety of trace gases can be detected. Here we compare both ion sources with respect to sensitivity and achievable detection limits of a number of trace gases. Figure 3 illustrates the primary-ion spectrum with discharge ion source under various conditions and compares it to that obtained using $^{210}$Po. The absolute ion count rates for both ion sources are comparable (in the order of $10^6$ to $10^7$ Hz for I$^-$). Details of the configurations (i-v) and primary-ions observed are summarised in Table 1.

The primary-ion mass spectrum obtained by passing CH$_3$I / N$_2$ through the $^{210}$Po-ioniser (370 MBq, configuration i) at typical relative humidity (50 % at 25 °C) and low de-clustering voltage (0–2 V) in the CDC is dominated by I$^-$ and I$^-$(H$_2$O) at $m/z$ 127 and 145, with no other significant ion peaks present at > 0.1 % relative signal strength to I$^-$. The background signal



for all trace gases of interest, i.e. PAN and PAA at $m/z$ 59 and $ClNO_2$ at $m/z$ 208 and 210 is consequently negligible and the detection limits are correspondingly low (a few pptv in 1 s integration time).

With RF discharge and $CH_3I$ / $N_2$ as ion source gas (configuration iii, applied in all the field measurements we discuss later) the primary-ion mass spectrum is more complex with additional ions such as $CNO^-$ ($m/z$ 42, 37% of $I^-$ at low de-clustering),

$CO_3^-$ ($m/z$ 60, 32% of $I^-$), $NO_3^-$ ($m/z$ 62, 5% of $I^-$), $IO_3^-$ ($m/z$ 175, 28% of $I^-$) and $I(CN)_2^-$ ($m/z$ 179, 67% of $I^-$). With the de-clustering voltage set to 20 V (best S/N ratio for most molecules of interest), the background count rate compared with $^{210}Po$ is elevated by at least one order of magnitude for $m/z$ 59 which is used to monitor PAN and PAA. The high chemical background is assumed to originate from $CH_3I$ breakdown in the discharge ion source and formation of $O_2^-$ in the ion source and IMR. When pumping the region around the ion source needles the formation of $NO_3^-$ ($m/z$ 62) can be reduced by a factor

of 2 but the other ions still show similar ion count rates. This observation suggests that a small amount of $O_2$ entering the ion source can form additional $NO_3^-$. According to manufacturer's specifications the nitrogen supply ($N_2$ 6.0, Westfalen) can contain up to 0.5 ppmv $O_2$ and $H_2O$ that can also result in formation of $NO_3^-$ in the ion source.

To examine the influence of oxygen in the IMR on the primary-ions formed we switched the main gas-flow (i.e. that which does not pass through the RF discharge) from ambient air to pure nitrogen (configuration iv). In this case, apart from $I^-$, only

$CN^-$ ($m/z$ 26, 2% of $I^-$ at low de-clustering), $IH(CN)^-$ ($m/z$ 154, 6% of $I^-$) and $I(CN)_2^-$ ($m/z$ 179, 9% of $I^-$) remained. While these conditions are unrealistic for atmospheric measurements they clearly indicate that the presence of additional primary ions containing O-atoms and the elevated chemical background on $m/z$ of interest are highly dependent on the amount of $O_2$ present in the IMR. With pure $N_2$ in the inlet (configuration iv), background signals at $m/z$ 59, 188, 207 and 208 could be lowered by about an order of magnitude. The use of $N_2$ results in a drastically reduced sensitivity to $SO_2$, as the primary ion

used to detect it ($IO_3^-$, see Sect. 4.2) is no longer abundant. This is illustrated in Fig. S2 of the supplementary information where we plot the dependence of the $IO_3^-$ signal ($m/z$ 175) on the fractional pressure of $O_2$ in the inlet and the signal at $m/z$ 207 ($ISO_3^-$) used to monitor $SO_2$ (see Sect. 4.2). Clearly, detection of $SO_2$ is not possible without the presence of $O_2$ and is not available when using $^{210}Po$ as ion source.

In an attempt to improve the detection limit for PAN by lowering the background signal on $m/z$ 59, $I_2$, produced from a flow

of nitrogen over iodine crystals (configuration v), was used instead of $CH_3I$. This resulted in a significant reduction of background signals, especially on $m/z$ 59 and the disappearance of all the ion peaks containing C and N atoms with just $IO_3^-$ ($m/z$ 175, 51% of $I^-$), $IO_4^-$ ($m/z$ 191, 38% of $I^-$), $IO_2^-$ ($m/z$ 159, 12% of $I^-$) and $NO_3^-$ ($m/z$ 62, 8% of $I^-$) remaining. Use of $I_2$ was however accompanied by a drastic lowering of the sensitivity to PAN, despite comparable $I^-$ ion counts at $m/z$ 127. The decrease in sensitivity can be traced back to equilibrium between $I^-$, $I_2$ and $I_3^-$.

30         $I^- + I_2 + M \quad\longleftrightarrow\quad I_3^- + M$                                         (R7)

An equilibrium constant ($K_{eq}$, in $bar^{-1}$) of $K_{eq} = [I_3^-]/[I^-][I_2] = \exp(11300/T)$ for reaction R7 (based on the Gibbs free energy of -94.14 kJ $mol^{-1}$ (NIST Webbook, 2010) and an estimate (based on its saturation vapour pressure) of the concentration of $I_2$ of $\approx 2 \times 10^{-7}$ bar) results in the complete dominance (by several orders of magnitude) of $[I_3^-]$ compared to $[I^-]$ in the IMR.





While the presence of large concentrations of $I_3^-$ may explain the large signal at *m/z* 127 following de-clustering, we conclude that the reaction between $I_3^-$ and $CH_3C(O)O_2$ is very inefficient or does not lead to $CH_3CO_2^-$ formation. $I_2$ does not represent a feasible alternative to $CH_3I$ for PAN measurement and HCl detection is not possible. However, we can still detect $SO_2$ via $IO_x^-$ primary-ions and also acetic acid presumably due to $(IO_x^-)$-clusters with $CH_3C(O)OH$.

In another experiment performed with the [210]Po source, synthetic air instead of nitrogen was flowing over the polonium ioniser (configuration ii), simulating a huge leak of oxygen into the source. Besides $I^-$ and $I^-(H_2O)$, $O_2^-$ (*m/z* 32, 5% of $I^-$), $O_2^-$ $(H_2O)$ (*m/z* 50, 3% of $I^-$) and $CO_3^-$ (*m/z* 60, 2% of $I^-$) were present but ions like $IO_X^-$ and $I(CN)_X^-$ that are probably responsible for the detection of $SO_2$ and HCl (see later) were not observed as they are unique to the RF discharge ion source. In combination with the experiment where the housing around the tungsten needles was evacuated, we conclude that a leak

of $O_2$ into the discharge region might increase $O_2^-$ and $NO_3^-$ but is not responsible for the complex primary-ion spectrum observed with our discharge ion source. No change in the ion spectrum was observed when the linear steel-tubing between ion source and IMR was replaced by tubing with a 90° bend. This result precludes an important role for ion formation via highly energetic radiation from the ionisation region reaching the IMR interacting with $O_2$. The diffusion of oxygen from the IMR into the ion source is also very unlikely due to a high-volume flow between ion source and IMR and the use of a small

aperture. We conclude that the role of $O_2$ in formation of primary ions containing I, O, C and N atoms in the IMR is most likely related to its role as trapper and carrier of electrons, possibly as excited $O_2^-$ anions.

## 4 Detection schemes and calibration methods

Figure 4 shows which trace gases we can detect with our instrument using the RF discharge ion source. The middle branch (b) represents ionisation via $I^-$ and its water cluster ion and is the same as [210]Po based ion generation schemes frequently used

for detection of PAN (Slusher et al., 2004), PAA (Phillips et al., 2013) and $ClNO_2$ (McNeill et al., 2006). The outer branches (a and c) are unique to our CI-QMS using the discharge ion source and can be attributed to the presence of different primary ions. The presence of H, C and N atoms from $CH_3I$ / $N_2$ breakdown in the discharge region leads to the existence of the right-hand branch (c) that disappears when $CH_3I$ is replaced by $I_2$ (see Sect. 3, configuration v). The usual presence of oxygen inside the IMR is responsible for the existence of the left-hand branch (a) and is not available with configuration (iv)

in which nitrogen was used for the main gas flow instead of synthetic air. For the combined detection of PAN, $ClNO_2$, $SO_2$, HCl, peracetic and acetic acid, configuration (iii) was used in all our field measurements. In the following we discuss in detail the ion-molecule-reactions involved in the detection of these trace gases and also outline how the CI-QMS is calibrated and which sensitivities and detection limits can be achieved.

### 4.1 PAN, PAA and acetic acid

PAN, PAA and acetic acid are all detected as $CH_3CO_2^-$ at *m/z* 59. Depending on relative humidity and the de-clustering potentials in the CDC, clusters of $CH_3CO_2^-$ with $H_2O$ are observed at *m/z* 77. Sensitivities and product yields for the detection of these three molecules are summarised in Table 2. The detection mechanism for PAN using $I^-$ primary-ions is the



same as that reported when using $^{210}$Po as ion source (Slusher et al., 2004). PAN is thermally decomposed inside the TDR into a peroxy radical ($CH_3C(O)O_2$) and $NO_2$ via Reaction (R8). The $CH_3C(O)O_2$ product reacts with $I^-$ in the IMR to form $CH_3CO_2^-$ (*m/z* 59) via Reaction (R9) involving clusters with water vapour. The detection of PAA with $I^-$ primary ions is believed to be direct, via Reaction (R10) (Phillips et al., 2013).

$$CH_3C(O)O_2NO_2 + M \quad \rightarrow \quad CH_3C(O)O_2 + NO_2 + M \tag{R8}$$

$$CH_3C(O)O_2 + I^-(H_2O)_n \quad \rightarrow \quad CH_3C(O)O^-(H_2O)_n + IO \tag{R9}$$

$$CH_3C(O)OOH + I^-(H_2O)_n \quad \rightarrow \quad CH_3C(O)O^-(H_2O)_n + HOI \tag{R10}$$

When using the RF discharge ion source there is also an additional pathway for PAN and PAA detection, involving $I(CN)_2^-$ primary ions, resulting in formation of $I(CN)CH_3CO_2^-$ which is observed at *m/z* 212 when de-clustering is switched off. With

de-clustering, this ion fragments to *m/z* 59. However, the sensitivity is relatively low and the selectivity not improved as acetic acid is also detected at this *m/z* (see Table 2).

The separation of PAN from PAA / acetic acid signals when sampling air masses which contain both trace gases can be achieved by cooling the TDR to prevent formation of $CH_3C(O)O_2$ and thus detection of PAN (Phillips et al., 2013) or by adding NO to the TDR in order to remove $CH_3C(O)O_2$ (Reaction R11).

$$CH_3C(O)O_2 + NO \quad \rightarrow \quad CH_3 + CO_2 + NO_2 \tag{R11}$$

The latter method has the advantage of being more rapid as NO can be switched in and out of the TDR in a matter of seconds whereas cooling of the inlet may take minutes. Generally we add NO (100 ppmv in nitrogen, Air Liquide) to the TDR at a mixing ratio of 0.23 ppmv ($3.7 \times 10^{12}$ molecule $cm^{-3}$). Given the approximate residence time of circa 200 ms in the TDR (calculated from the volume of the TDR and the volumetric flow rate) and the rate coefficient for Reaction (R11) of $1.4 \times$

$10^{-11}$ $cm^3$ molecule$^{-1}$ s$^{-1}$ at 453 K (Atkinson et al., 2004; IUPAC, 2018) we calculate that more than 99.99 % of the $CH_3C(O)O_2$ is removed by titration with NO. When NO is added, we therefore measure only PAA (or the sum of PAA and acetic acid, depending on de-clustering potentials, see below). The signal due to PAN is then calculated by subtracting the interpolated signal during NO addition.

The use of the RF discharge source also results in sensitivity to acetic acid at *m/z* 59, which is not observed using $^{210}$Po as ion

source. Rather, detection of acetic acid has been reported at *m/z* 187 ($ICH_3C(O)OH^-$) (Lee et al., 2014). Our instrument is relatively insensitive for acetic acid at *m/z* 187 and, without de-clustering, we measure acetic acid mainly as $(CNO)CH_3C(O)OH^-$ (*m/z* 102) and $I(CN)CH_3CO_2^-$ (*m/z* 212).

When monitoring *m/z* 59 using the discharge ion source, the background signal during NO addition consists of both peracetic (PAA) and acetic acid. To differentiate between them we make use of the fact that the relative sensitivity to PAA and acetic

acid at *m/z* 59 depends on the de-clustering potentials applied in the CDC. We find that acetic acid is only detected at *m/z* 59 when a de-clustering voltage of about 20 V is applied, whereas PAA is detected at *m/z* 59 both with and without de-clustering, albeit with different sensitivity. The difference is related to the fact that for PAA the product ion is formed directly in Reaction (R10), whereas acetic acid is believed to initially form a cluster with $I(CN)_2^-$ (Reaction R12) and only dissociate to $CH_3CO_2^-$ when the 20 V de-clustering voltage is applied (R13). $ICN^-$, $IH(CN)^-$ and $I(CN)_2^-$ are all potential



primary ions for detection of acetic acid, though $I(CN)_2^-$ is the most abundant. In principal, the mixing ratio of acetic acid can be calculated by subtracting the signal without de-clustering from the signal with de-clustering. Unfortunately, the chemical background without de-clustering at $m/z$ 59 is by a factor of 2.5 higher and the sensitivity relatively low which increases the LOD for detection of PAA significantly (see Table 2).

$$CH_3C(O)OH + I(CN)_2^- \quad \rightarrow \quad HCN + I(CN)CH_3CO_2^- \quad (m/z\ 212) \quad\quad (R12)$$

$$I(CN)CH_3CO_2^- + \Delta U \quad \rightarrow \quad ICN + CH_3CO_2^- \quad (m/z\ 59) \quad\quad (R13)$$

Results of combined PAA and acetic acid measurements from the CYPHEX field campaign and speciation via changing the CDC parameters can be found in Derstroff et al. (2017).

An enhancement in sensitivity to acetic acid at $m/z$ 59 when using the RF discharge ion source compared to $^{210}$Po, is
illustrated by the signals obtained from the photochemical source used to generate PAN, which also generates unquantified amounts of both PAA and acetic acid (Sect. 2.7, Reactions R5 and R6). When using the RF discharge ion source and 20 V de-clustering voltage, the signal-ratio PAN / (PAA + acetic acid) at $m/z$ 59 is $\approx$ 0.2. In contrast, using $^{210}$Po as ion source the PAN / (PAA + acetic acid) ratio is circa 0.9. As the relative sensitivity ($m/z$ 59) to PAN and PAA is similar, this change in ratio reflects enhanced instrument sensitivity to acetic acid when using the discharge ion source with de-clustering. This
represents a significant disadvantage of the RF discharge source for PAN detection compared to $^{210}$Po. Not only is the instrumental background at $m/z$ 59 higher, the presence of acetic acid in ambient air samples means that a larger and more variable chemical background signal has to be subtracted to calculate the PAN mixing ratio, which increases the limit of detection and overall uncertainty significantly.

### 4.1.1 Calibration of PAN, PAA and acetic acid

The in-situ calibration of PAN is described in Sect. 2.7. The overall uncertainty of the calibration, based on the uncertainty in dilution, the mixing ratio of the NO calibration cylinder (1 ppmv) and the conversion efficiency from NO to PAN is $\approx$ 10 %. For PAA, two methods, both using a diffusion source containing a commercially available 39% solution of PAA in acetic acid, have been used to calibrate the CI-QMS. In the first, we use simultaneous CI-QMS and wet-chemical peroxide-specific detection of PAA based on the horseradish peroxidase / catalase / p-hydroxyphenyl wet chemical fluorescence measurement
technique (Lazrus et al., 1986) in which organic peroxides (and per-acids) are converted to $H_2O_2$ (Model AL2021, Aero-Laser GmbH). The wet-chemical method is calibrated via standard $H_2O_2$ solutions and the overall uncertainty (related to scrubbing efficiency of PAA) is 13%. As the AL2021 is not always available during campaign, we developed a second approach in which PAA undergoes wet chemical transformation to $I_3^-$ (aq), which can be quantified using aqueous-phase absorption spectroscopy (Awtrey and Connick, 1951; Friedrich, 2015). Based on uncertainty in the scavenging of PAA into
acidified, aqueous solution, uncertainty associated with the absorption cross-section of $I_3^-$ and the reproducibility of $I_3^-$ signals when sampling from a constant source of PAA, we estimate the total uncertainty of the $I_3^-$ method to be $\approx$ 30%.





For the calibration of acetic acid we use a permeation source (7.33 ng min$^{-1}$ at 30 °C, Metronics) with an uncertainty of 8%. Additionally, we sampled the output of a diffusion source of pure liquid acetic acid simultaneously using the CI-QMS (41.2 ppbv after dilution) and an infrared absorption spectrometer measuring $CO_2$ following the stoichiometric, thermal oxidation of acetic acid to $CO_2$ (LICOR). The uncertainty of this calibration method is ≈ 10%. Within combined uncertainty, both

methods indicated the same sensitivity of the CI-QMS to acetic acid.

**4.2. Sulphur dioxide**

The "standard" use of $^{210}$Po-ionisation does not allow for the sensitive detection of $SO_2$ using I$^-$ ions. As outlined in Sec. 3 additional primary ions ($IO_x^-$) formed with our RF discharge ion source enable $SO_2$ detection as e.g. $ISO_3^-$ (*m/z* 207). In addition, $ISO_4^-$, $SO_4^-$ and $SO_5^-$ are also formed and the relative yields are listed in Table 2. Although the underlying ion-

molecule-reactions resulting in their formation are not fully characterised, based on the observation of $IO_3^-$ and $IO_4^-$ in the primary-ion mass spectrum we propose the following scheme (Reactions R14–17).

| | | | | |
|---|---|---|---|---|
| $SO_2 + IO_3^-$ | $\rightarrow$ | $O_2 + ISO_3^-$ | (*m/z* 207) | (R14) |
| $SO_2 + IO_4^-$ | $\rightarrow$ | $O_2 + ISO_4^-$ | (*m/z* 223) | (R15) |
| $SO_2 + IO_3^-$ | $\rightarrow$ | $IO + SO_4^-$ | (*m/z* 96) | (R16) |
| $SO_2 + IO_4^-$ | $\rightarrow$ | $IO + SO_5^-$ | (*m/z* 112) | (R17) |

As written, reactions (R14) and (R16) involving the $IO_3^-$ anion are exothermic with reaction enthalpies of ≈ -250 and -113 kJ mol$^{-1}$, respectively. The enthalpies of formation used to derive these value were taken from the literature: $\Delta H_f^{298}(SO_2) =$ -287 kJ mol$^{-1}$ (Chase, 1998), $\Delta H_f^{298}(IO) = 126$ kJ mol$^{-1}$ (Goos et al., 2005), and $\Delta H_f^{298}(SO_4^-) = -738$ kJ mol$^{-1}$ ((NIST Webbook, 2010) or calculated from other thermodynamic properties. The formation enthalpy for $IO_3^-$ ($\Delta H_f^{298}(IO_3^-) = -211$ kJ

mol$^{-1}$) was calculated from its electron affinity (453.5 kJ mol$^{-1}$, (Wen et al., 2011) and the formation enthalpy of $IO_3$ (242 kJ mol$^{-1}$, (Goos et al., 2005)). The formation enthalpy for $ISO_3^-$ ($\Delta H_f^{298}(ISO_3^-) = -752$ kJ mol$^{-1}$) was calculated from the $SO_3$-I$^-$ bond strength (161 kJ mol$^{-1}$, (Hao et al., 2005)) and the formation enthalpies of I$^-$ (-195 kJmol$^{-1}$, (Goos et al., 2005)) and $SO_3$ (-396 kJ mol$^{-1}$, (Goos et al., 2005)). In the absence of thermodynamic data for $IO_4^-$, we cannot assess the reaction enthalpies for Reactions (R15) and (R17). The iodine containing $ISO_3^-$ (*m/z* 207) is most specific and best suitable for monitoring $SO_2$

with good sensitivity.

**4.2.1 Calibration of SO₂**

$SO_2$ is calibrated by addition of a small flow of $SO_2$ from a gas cylinder (1 ppmv in synthetic air, Air Liquide). The true mixing ratio of $SO_2$ flowing from the bottle into the absorption cell at 1 bar pressure was determined via UV absorption

spectroscopy using a white-cell / diode array set up (Wollenhaupt et al., 2000) and an absorption spectrum (290–320 nm) from the literature (Bogumil et al., 2003). The mixing ratio thus determined agreed to within 10 % of the manufacturers





specifications. The linearity of the CI-QMS signal with $SO_2$ mixing ratio (up to 60 ppbv) is shown as Fig. S3a in the supplementary information.

### 4.3 Nitryl chloride

The scheme for detection of $ClNO_2$ using $I^-$ ions generated using $^{210}Po$ is well established (Osthoff et al., 2008; Thornton et al., 2010). Both $ICl^-$ (*m/z* 162 and 164) and $IClNO_2^-$ (*m/z* 208 and 210) are formed, the latter generally preferred to monitor $ClNO_2$ in ambient air owing to potential interference through reactions of other chlorine containing trace gases forming $ICl^-$.

| | | | |
|---|---|---|---|
| $ClNO_2 + I^-$ | → | $IClNO_2^-$ | (R18) |
| $ClNO_2 + I^-$ | → | $ICl^- + NO_2$ | (R19) |

The same ions are observed during operation with the RF discharge ion source with similar product yields for $^{210}Po$ as the ones for the RF discharge reported in Table 2.

#### 4.3.1 Calibration of $ClNO_2$

$ClNO_2$ was calibrated by passing $Cl_2$ (50 ppmv in nitrogen, Air Liquide) over $NaNO_2$ (30 g) and $NaCl$ (10 g) crystals in a glass flask (Thaler et al., 2011). The $ClNO_2$ generation efficiency was found to be improved by moistening the crystals by adding 2–3 drops of water. The gas-mixture exiting the glass flask, which contains unreacted $Cl_2$ and $NO_2$ as well as $ClNO_2$ was diluted in 5 slm air and sampled simultaneously with the CI-QMS and a cavity ring-down spectrometer that detects both $NO_2$ and $ClNO_2$ after thermal decomposition at 420 °C to $NO_2$ (Thieser et al., 2016). $ClNO_2$ thermograms (Sobanski et al., 2016a; Thieser et al., 2016) indicate that, under the flow and pressure conditions of these calibrations the $ClNO_2$ is thermally decomposed to $NO_2$ at 420 °C (Reaction R20) but there is no significant loss at 200 °C, the TDR temperature of the CI-QMS.

| | | | |
|---|---|---|---|
| $ClNO_2 + M$ (420 °C) | → | $Cl + NO_2 + M$ | (R20) |

The total uncertainty ($\approx$ 25 %) associated with the calibration derives from uncertainty in the $NO_2$ cross-section used to calculate $NO_2$ mixing ratios in the cavity-ring-down spectrometer and the assumption that all $ClNO_2$ is detected as $NO_2$.

### 4.4 Hydrogen chloride

Using the discharge ion source we detect HCl as $Cl^-$, $ICl^-$ and $I(CN)Cl^-$, presumably via Reaction (R21–23):

| | | | | |
|---|---|---|---|---|
| $HCl + I(CN)_2^-$ | → | $HCN + I(CN)Cl^-$ | (*m/z* 188 and 190) | (R21) |
| $HCl + ICN^-$ | → | $HCN + ICl^-$ | (*m/z* 162 and 164) | (R22) |
| $HCl + CN^-$ | → | $HCN + Cl^-$ | (*m/z* 35 and 37) | (R23) |

Reactions (R22) and (R23) are weakly exothermic with reaction enthalpies of $\approx$ -25 and -73 kJ $mol^{-1}$, respectively. If available, the enthalpies of formation used to derive these values were taken from Goos et al. (2005): $\Delta H_f^{298}(HCl)$ = -92 kJ $mol^{-1}$, $\Delta H_f^{298}(HCN)$ = 130 kJ $mol^{-1}$, $\Delta H_f^{298}(Cl^-)$ = -234 kJ $mol^{-1}$ and $\Delta H_f^{298}(CN^-)$ = 61 kJ $mol^{-1}$ and Refaey and Franklin (1977):





$\Delta H_f^{298}(ICl^-) = -155$ kJ mol$^{-1}$. $\Delta H_f^{298}(ICN^-) = 92$ kJ mol$^{-1}$ was calculated from $\Delta H_f^{298}(ICN) = 222$ kJ mol$^{-1}$ (Goos et al., 2005) and the electron affinity of ICl (130 kJ mol$^{-1}$, (Miller et al., 2012)). The I(CN)$_2^-$ ion is stable in aqueous solution (Chadwick et al., 1980) but lacking thermodynamic data for it and for I(CN)Cl$^-$ preclude calculation of the energetics of Reaction (R21).

As I(CN)Cl$^-$, formed by reaction of dicyano-iodate anion with HCl in Reaction (R21), is the most abundant and specific product ion, HCl is generally monitored at $m/z$ 188. In the absence of interferences, the ratio of signals at $m/z$ 188 to $m/z$ 190 and $m/z$ 162 to $m/z$ 164 should be determined by the natural, relative abundance of the $^{35}$Cl and $^{37}$Cl isotopes which is $\approx$ 3.13. Plots of the relative ion signals at $m/z$ 188 versus $m/z$ 190 and $m/z$ 162 versus $m/z$ 164 obtained during the CYPHEX campaign are given in Fig. S4 of the supplementary information. The tight correlation and the slope of 3.09 for the ratio

$m/z$ 162 to $m/z$ 164 is very close to the expected value, indicating that to a good approximation, only one trace gas containing one Cl-atom was measured. In contrast, the ratio of $m/z$ 188 to $m/z$ 190 is significantly lower than expected, and the correlation displays more scatter. This low ratio indicates that $m/z$ 190 suffers from interference from another trace gas. A likely but unproven candidate is HNO$_3$, detected as IHNO$_3^-$ at $m/z$ 190. The signals at $m/z$ 188 and $m/z$ 162 are correlated very well (R$^2$ = 0.96) indicating that they both represent HCl only, as no significant ClNO$_2$ was present during CYPHEX.

### 4.4.1 Calibration of HCl

A bottle of gaseous HCl diluted in N$_2$ (60 ppmv) was used to calibrate the CI-QMS during laboratory operation. The concentration of HCl was determined using UV absorption spectroscopy (184.95 nm) using a cross section of $2.39 \times 10^{-19}$ cm$^2$ molecule$^{-1}$ (Bahou et al., 2001) as described by Zimmermann et al. (2016). Once the sensitivity of the CI-QMS to HCl

was established using bottled gas, the output of a laboratory-built permeation source was measured by comparing signals in the CI-QMS at $m/z$ 188. The permeation source consisted of a few ml of concentrated HCl-solution welded into a short length (4 cm) of ¼ inch PFA tubing, housed in 20 cm of ½ inch PFA tubing (at 30 °C) through which 50 sccm of air flows. The permeation rate measured was $5.2 \times 10^{-5}$ sccm with an uncertainty of the HCl calibration of $\approx$ 10%. The linearity of the CI-QMS signal with HCl mixing ratio was characterised in the laboratory (R$^2$ = 0.99) and is shown as Fig. S3b in the

supplementary information.

### 5 Dependence of sensitivity on relative humidity

The CI-QMS sensitivity for the trace gases discussed here is dependent on the amount of water vapour present in the IMR, which will vary with ambient relative humidity (RH). Broadly speaking, we observe a positive dependence of the sensitivity (see Fig. 5) on relative humidity between 0 and 20 % at 25 °C, with a flattening of the curve between 20 and 80 % RH. This

effect is generally explained by the reactions proceeding predominantly through clustered primary ions, e.g. I$^-$(H$_2$O) which is observed at $m/z$ 145. Under weak de-clustering conditions, the product ions are also clustered with H$_2$O, confirming the participation of I$^-$(H$_2$O). For the ISO$_3^-$ and I(CN)$_2^-$ primary-ions used to detect SO$_2$ and HCl, the water clusters IO$_3^-$(H$_2$O)$_n$



and $I(CN)_2^-(H_2O)_n$ are not observed or are very weak, even without de-clustering. We observe that the concentration of $ISO_3^-$ and $I(CN)_2^-$ in the primary ion-spectra are however dependent on the presence of $H_2O$, which explains the RH dependence of the sensitivity of detection for $SO_2$ and HCl. All ambient measurements of the trace gases discussed here are therefore corrected for RH effects using calibration curves based on data such as that displayed in Fig. 5.

## 6 Sensitivity, detection limits and total uncertainty

The total uncertainty of the measurement of any of the trace gases listed above is determined mainly by the uncertainty associated with the calibration method (and its reproducibility) but may also be influenced by e.g. scrubbing efficiency and drifts between background measurements (variable for different field campaigns). The response of the CI-QMS to any one trace gas may also vary over a period of days to few weeks due to drifts in temperature, resolution of the mass spectrometer and degradation of the detector. The sensitivity (i.e. signal in Hz per pptv of trace gas) depends on the rate coefficient for reaction between primary-ion and trace gas and the yield of product ions. The sensitivity may also depend on relative humidity (abundance of $H_2O$ clusters) and de-clustering potential (breakup of weak bonds). The limit of detection (LOD) is mainly dependent on variability in the background signal on the respective $m/z$ and can be calculated as two times the standard deviation when using synthetic (i.e. hydrocarbon-free) air. In the text below, and summarised in Table 2, we report sensitivities (in Hz pptv$^{-1}$) and limits of detection (LOD, $2\sigma$, in pptv) obtained under typical measurement conditions (configuration iii from Sect. 3) and, when applicable, compare them to values obtained using $^{210}$Po as ion source.

### 6.1 PAN, PAA and acetic acid

When using $^{210}$Po as ion source, an LOD of 3 pptv PAN in 1 s is achievable, which is adequate for e.g. airborne operation (Roiger et al., 2011) or flux measurements (Wolfe et al., 2009). The use of the RF discharge for PAN detection results in an increase in background signal (from a few Hz when using $^{210}$Po to several hundred Hz when using the RF discharge ion source) even in hydrocarbon-free, synthetic air. The LOD calculated from twice the standard deviation of a background measurement during the NOTOMO campaign is 34 pptv in 1 s. The total uncertainty calculated from measurement precision, background subtraction (signal drifts, interpolation) and uncertainty in the calibration method is 15 % +/- 27 pptv. However, the uncertainty of the PAN measurement is highly dependent on the levels and variability of PAA and acetic acid present in the air as their signal has to be interpolated and subtracted. In ambient air masses, the larger part of the signal at $m/z$ 59 with the RF discharge is due to acetic acid which sometimes displays variability on the time-scale of minutes. In this case the uncertainty of the background interpolation and therefore the overall uncertainty of the PAN measurement are drastically increased. This is essentially a selectivity problem which limits deployment of the instrument for PAN measurements to more polluted regions where PAN mixing ratios regularly exceed hundred pptv and/or high time resolution is not necessary. As described above, the selectivity to PAN can be improved by switching off de-clustering (no acetic acid detection at $m/z$ 59), which however comes with a significant reduction in sensitivity (see below).



In order to differentiate between PAA and acetic acid, the de-clustering voltage has to be modulated between 20 V and 2 V. At the lower voltage, only PAA is detected but the resultant high chemical background and worsened sensitivity were found to lead to a poor limit of detection of a few hundred pptv in 1 s (see Table 2) which is about a factor of 100 higher than with use of $^{210}$Po (4 pptv in 1 s). At the higher potential the LOD would be much better but the sensitivity to acetic acid at $m/z$ 59 reduces the selectivity of the measurement. The total uncertainty calculated from measurement precision, background subtraction (signal drifts, interpolation) and uncertainty in the calibration method is 20 % ± 39 pptv.

The LOD for acetic acid at $m/z$ 59 is 57 pptv in 1 s but the selectivity reduced due to the PAA contribution. The total uncertainty calculated from measurement precision, background subtraction (signal drifts, interpolation) and uncertainty in the calibration method is 15 % ± 45 pptv.

## 6.2 SO₂

The sensitivity of the CI-QMS to SO₂ reported in Table 2 is dependent on the relative humidity (see Sect. 5) and especially on the de-clustering voltage, the best signal-to-noise ratio being found at 20 V. Although $HSO_4^-$ ($m/z$ 97) has the highest sensitivity of all product ions, we generally monitor the $ISO_3^-$ ion ($m/z$ 207) as the background signal is lower and the detection limit improved. Figure S5 of the supplementary information displays the correlation between both $m/z$ over a period of four weeks during the NOTOMO campaign. The correlation coefficient ($R^2$ = 0.95) is large, from which we conclude that both $m/z$ can be used to calculate SO₂ mixing ratios. The detection limits for $m/z$ 207 is 56 pptv in 1 s (based on noise in background measurements during NOTOMO field campaign) which is sufficient to monitor SO₂ in lightly polluted areas. At lower temporal resolution and when monitoring only $ISO_3^-$ and $I^-$ the LOD can be improved to a few pptv (e.g. in 10 min). The total uncertainty, calculated from measurement precision, background subtraction (signal drifts, interpolation) and uncertainty in the calibration method is 20 % ± 23 pptv.

## 6.3 ClNO₂

Very good detection limits have been reported (Osthoff et al., 2008; Thornton et al., 2010; Phillips et al., 2012) for the measurement of ClNO₂ via I-CIMS using $^{210}$Po-ionisation, a result of low background signal at $m/z$ 208 ($IClNO_2^-$) and an efficient reaction with $I^-$. Using $^{210}$Po, Phillips et al. (2012) achieved a LOD (2σ) of 3 pptv in 1 s which can be compared to the value of 12 pptv in 1 s (see Table 2) obtained with the RF discharge ion source, the difference stemming from a higher chemical background signal. With an averaging interval of 5 min the LOD can be reduced to 2–3 pptv. ClNO₂ can also be detected as $ICl^-$ ($m/z$ 162 and 164) which provides higher sensitivity compared to $IClNO_2^-$ (see Table 2) but can suffer from a significant interference due to HCl which is likely to be present in air masses containing ClNO₂. For example, 1 ppbv HCl contributes a signal at $m/z$ 162 which is equivalent to 60 pptv ClNO₂ at this $m/z$. Monitoring ClNO₂ at $m/z$ 208 is more specific, with an equivalent signal due to 1 ppb HCl of less than 10 pptv, which can be accounted for when measuring HCl in parallel (at $m/z$ 188, see above). It should be noted that the interference at $m/z$ 162 is not unique to the RF discharge but has



also been observed when using a $^{210}$Po-ioniser (Phillips et al., 2012). The total uncertainty for ClNO$_2$ measurement, calculated from precision, background subtraction (signal drifts, interpolation) and uncertainty in the calibration method is 30 % $\pm$ 6 pptv.

### 6.4 HCl

Sensitivities and product yields for several ions connected to HCl detection are reported in Table 2. As *m/z* 162 and 164 (ICl$^-$) suffer from a ClNO$_2$ interference (see above) and Cl$^-$ (*m/z* 35 and 37) could possibly arise from other Cl-containing species, the more specific ion I(CN)Cl$^-$ (*m/z* 188 and 190) is used to monitor HCl. The LOD for *m/z* 188 is 135 pptv in 1 s, which can be further improved by extended averaging if high time resolution is not required. The total uncertainty calculated from measurement precision, background subtraction (signal drifts, interpolation), scrubbing efficiency (it takes more time to remove HCl in the scrubber than e.g. SO$_2$) and uncertainty in the calibration method is 20 % $\pm$ 72 pptv.

## 7 Application in the field

Our CI-QMS instrument has been deployed in different ground-based field campaigns including coastal (CYPHEX, 2014), forested (IBAIRN, 2016) and mountain sites (NOTOMO, 2015) in Europe. In the following we present sub-sets of the data from these campaigns in order to indicate how the instrument with RF discharge ion source (configuration iii in Sect. 3) performs in the field.

### 7.1 CYPHEX 2014

During CYPHEX ("Cyprus Photochemistry Experiment", summer 2014), located at a coastal site on the eastern Mediterranean island of Cyprus, we measured chemically aged air masses with origin in continental Europe (Meusel et al., 2016; Derstroff et al., 2017). A time series of SO$_2$ and HCl for a 3 weeks period of the campaign is shown in Fig. 6. SO$_2$ was detected for the first time using the CI-QMS during CYPHEX in which observations of co-variance between the signal at *m/z* 207 and particulate sulphate provided the first clues to the identity of the mass peak as ISO$_3^-$ and indications of sensitivity to SO$_2$. As we had not anticipated CI-QMS sensitivity to SO$_2$, calibration was performed post-campaign. We observed SO$_2$ mixing ratios as high as 11 ppbv, the plume like nature of which strongly suggest nearby point-sources such as ship traffic or air masses originating from power-plants in continental Europe. Our measurements are consistent with other observations in the coastal Mediterranean boundary layer. Bardouki et al. (2003) found SO$_2$ mixing ratios up to 3 ppbv in Crete (August 2001) and Schembari et al. (2012) report average daily mean values of several ppbv in different western Mediterranean harbours measured in summer 2009 and 2010. Kanakidou et al. (2011) conclude that megacities can be hotspots of air pollution in the East Mediterranean with average SO$_2$ mixing ratios of 1 ppbv measured in Crete (1997-1999), 8 ppbv in Istanbul (1998-2008), 10-15 ppbv in Athens (1995-1997) and 48 ppbv in Cairo (1999-2000) where about 70% originates from industrial activities.





Similar to $SO_2$, the measurement of HCl was unexpected, the isotopic ratio of 3:1 for the signals at $m/z$ 162 and 164 ($ICl^-$) providing evidence that the trace gas measured contained one Cl atom. The identification of HCl was confirmed during the campaign, in which a permeation source was built and used to periodically supply HCl to the CI-QMS. Calibration of the permeation source ensued post-campaign as described in Sec. 4.4.1. HCl mixing ratios up to 3 ppbv were observed which

could be attributed to the release of HCl from sea salt aerosols when polluted air masses from continental Europe reached the coastal site. The covariance between HCl and $SO_2$ in Fig. 6 suggests that an acid displacement mechanism involving $SO_2$ oxidation to $H_2SO_4$ and transfer of $H_2SO_4$ to aqueous sea salt aerosol was involved. The median value of the HCl mixing ratio for the CYPHEX campaign was 790 pptv which can be compared with reports of median values of 600–700 pptv HCl over the Atlantic near Europe and values of up to 6 ppbv in the polluted coastal boundary layer on the Isles of Shoals, 10 km

off the southern Maine coast, USA, with an average of 600 pptv (July–August 2004) (Keene et al., 2007; Keene et al., 2009).

## 7.2 NOTOMO 2015

The mountain-site campaign NOTOMO ("NOcturnal chemistry at the Taunus Observatory: insights into Mechanisms of Oxidation", summer 2015) took place in a rural location in south-western Germany with significant urban influence

(Sobanski et al., 2016b). A time series of $SO_2$, $ClNO_2$ and PAN mixing ratios and the signal at $m/z$ 59 (contributions from acetic and peracetic acid) is displayed in Fig. 7. During NOTOMO, $SO_2$ was monitored as $ISO_3^-$ ($m/z$ 207) and $HSO_4^-$ ($m/z$ 97) and a very good correlation between both signals ($R^2 = 0.95$) within a period of four weeks confirmed that both ions reliably represent the same molecule (see Fig. S5). The $SO_2$ mixing ratios exceeded 1 ppbv on most days, with maximum values up to 5 ppbv. The likely origins of $SO_2$ at this site are emissions from coal-burning power plants in the local Rhein-

Main urban conglomeration and the heavily industrialised Ruhr area to the North-West.

$ClNO_2$ was detected during NOTOMO as $IClNO_2^-$ ($m/z$ 208 and 210). Mixing ratios ranged from 0 to 500 pptv and were above 50 pptv during 10 out of 29 campaign nights. High levels of $ClNO_2$ were generally associated with mixed marine and continental air masses from the north-west which had passed over the English Channel and the polluted Ruhr area. The data is consistent with previous measurements (using the CI-QMS equipped with a $^{210}$Po ioniser) at the same location and similar

time of year (Phillips et al., 2012) whereby comparable $ClNO_2$ mixing ratios were observed (see Fig. S6 of the supplement). We measured PAN at $m/z$ 59 and observed mixing ratios generally ranging between 0 and 2 ppbv throughout NOTOMO, occasionally reaching 3 ppbv. PAN levels predominantly peaked in the afternoon where photochemical activity is usually highest. Compared with results from PARADE (see Fig. S6 of the supplement) in which PAN had been measured with a precursor version of this instrument with $^{210}$Po-ioniser, frequency and amplitude of the PAN mixing ratios throughout the

campaign were very similar. However, due to high and variable chemical background at $m/z$ 59, as already pointed out in Sect. 4.1, the detection limit during NOTOMO was about an order of magnitude worse than in PARADE (see Table 2). As we did not require higher temporal resolution than a few minutes for analysis, this was not a main issue here. The measurements with the RF discharge ion source had adequate sensitivity towards PAN at this moderately polluted region and have the additional advantage over $^{210}$Po of simultaneous detection of $SO_2$ and HCl. In contrast, the differentiation between





PAA and acetic acid is problematic. During NOTOMO, we were unaware of the sensitivity towards acetic acid at $m/z$ 59 and the signal without PAN (i.e. during titration of $CH_3C(O)O_2$ with NO) was measured with 20 V de-clustering only and therefore represents a combined signal of PAA and acetic acid. Although peracetic acid calibration was performed during the campaign, this is not considered reliable because the PAA diffusion source also contains significant amounts of acetic acid.

For this reason, we present only an upper limit for acetic acid, ranging between 0 and 8 ppbv. From the PAN-to-PAA ratio calculated for PARADE (about 10) we would also expect several hundreds of PAA to be present (see Fig. S6 of the supplement), which would lower this approximate acetic acid mixing ratio significantly.

### 7.3 IBAIRN 2016

The IBAIRN campaign ("Influence of Biosphere-Atmosphere Interactions on the Reactive Nitrogen budget", summer 2016) took place in the boreal forest in Hyytiälä, Finland, an area with large biogenic emissions and low-$NO_x$ conditions (Liebmann et al., 2018). The CI-QMS inlet was located at about 6 m height, within the canopy. A time series of $SO_2$, HCl, PAN, PAA and the signal at $m/z$ 59 having both contributions from acetic and peracetic acid is shown in Fig. 8.

During IBAIRN $SO_2$ was monitored by the CI-QMS as $ISO_3^-$ ($m/z$ 207) with mixing ratios up to 1 ppbv. $SO_2$ mixing ratios

were largest when the air originated from the North-East (point sources like coal-burning power plants in Northern Finland and Russia) but only occasionally exceeded 100 pptv in this remote, forested environment. Independent $SO_2$ measurements, using a TEI 43 CTL fluorescence analyser (SMEAR II station, University of Helsinki) were made on a tower at 16 m height (about 5 m distant from the CI-QMS inlet but approximately at canopy height) and allow a direct comparison to be made. The datasets are generally in good agreement (Fig. 8a), although some $SO_2$ plumes were only observed at the higher inlet

due to strong gradients in trace gas concentrations resulting from boundary layer dynamics.

HCl (Fig. 8b) was measured as $I(CN)Cl^-$ ($m/z$ 188 and 190) with mixing ratios up to 300 pptv showing a distinct diurnal profile with a maximum in the afternoon, which reflect temperature dependent changes in partitioning between the gas-phase and particle phase in which HCl is converted into $NH_4Cl$.

The combined PAA and acetic acid signal at $m/z$ 59 obtained with de-clustering at 20 V (Fig. 8e) displays the diurnal profile

expected from photo-chemically generated trace gases with night-time depositional losses. However, as explained in Sect. 7.2 we cannot easily separate the contribution to the signal from PAA and acetic acid. The signal at $m/z$ 59 with low de-clustering is due to PAA only (Fig. 8d), but PAA detection limit is poor due to a low sensitivity and elevated and variable background signal during zeroing. The PAA contribution to the total signal at $m/z$ 59 is likely to be substantial but cannot be calculated as the sensitivity of the CI-QMS to PAA during de-clustering (Fig. 8e) is unknown, a result of the presence of

unknown amounts of acetic acid in the PAA diffusion source. The use of a [210]Po-ioniser would have greatly improved the PAN (see Fig. 8c) and PAA data quality as evidenced by the PAN and PAA measurements reported for the same location during 2010 (Phillips et al., 2013; Crowley et al., 2018).





## 8 Conclusions

The CI-QMS with RF discharge ion source is a promising alternative to similar instruments using $^{210}$Po-based ion sources and can be deployed in environments for which permission to use $^{210}$Po may be difficult or impossible to obtain or to which transportation is not feasible. The use of the RF discharge results in an extension of the established detection schemes (e.g.

for ClNO$_2$ and PAN) using I$^-$ ions, and we have identified ion-schemes involving ISO$_X^-$ and I(CN)$_X^-$ primary-ions that additionally enable detection of SO$_2$ and HCl. Detection limits (2σ, 1 s integration) are 56 pptv for SO$_2$, 135 pptv for HCl and 12 pptv for ClNO$_2$ which makes the CI-QMS a useful tool for investigation of atmospheric processes related to sulphur and chlorine chemistry. Application of the instrument with RF discharge ion source for PAN detection is limited to polluted environments where mixing ratios usually exceed a hundred pptv and high temporal resolution is not needed. This restriction

is mainly due to a background count rate higher by 1-2 orders of magnitude compared to use of $^{210}$Po and due to a strong dependence of the measurement uncertainty on the variability of the subtracted interpolated background signal (consisting of PAA and acetic acid). A PAN detection limit (in the absence of PAA and acetic acid) of 34 pptv in 1 s was obtained, though this value will rarely be reached in boundary layer air masses where acetic acid and peracetic acid are abundant. Similarly, while sensitive detection of PAA (requiring de-clustering) is precluded by the detection of acetic acid at the same *m/z*, the

selective detection of acetic acid is uncertain due to the contribution of PAA.

The deployment of the CI-QMS with RF discharge and its advantages / disadvantages compared to instruments using $^{210}$Po-based ionisation are illustrated in three campaign datasets, which demonstrate its potential to monitor trace gases at mixing ratios ranging from a few tens of pptv to a few ppbv. If the science focus is on PAA and PAN, the RF discharge based CI-QMS is clearly disadvantaged compared to the more selective and sensitive $^{210}$Po-based ionisation. On the other hand, the

potential to measure ClNO$_2$ without logistic obstacles related to transport and (mobile) operation of radioactive sources and the added benefit of simultaneous measurement of HCl and SO$_2$ may in some instances tip the balance in its favour.

*Data availability.* The campaign datasets (times series for SO$_2$, ClNO$_2$, HCl, PAN etc.) used in this manuscript to illustrate the deployment of the CI-QMS can be obtained on request (via John N. Crowley) from the owners.

*Author contributions.* PGE developed and tested the RF discharge ion source in the laboratory, deployed the CI-QMS during the NOTOMO and IBAIRN campaigns, evaluated the field data and wrote the manuscript. FH and GS helped to design and modify the CI-QMS for constant pressure and RF discharge operation. JNC and JL designed the field campaigns and contributed to the manuscript, GJP operated the CI-QMS during CYPHEX and helped evaluate the data.

*Competing Interests.* The authors declare that they have no conflict of interest.

*Acknowledgements.* We thank the Cyprus Ministry of Defense for the use of the base of the National Guard at Ineia and the generous assistance of the Lara Naval Observatory staff during the CYPHEX campaign. Our thanks also go to the Department of Labor Inspection for helping us set up the campaign. We thank Heinz Bingemer for logistical support and use of the facilities at the Taunus Observatory during the NOTOMO campaign. We are grateful to the technical staff of SMEAR II station for technical support and ENVRIplus for partial financial support of the IBAIRN campaign. We are grateful for the

provision of automated SO$_2$ measurement data from the SMEAR II site in Hyytiälä: Junninen, H., Lauri, A., Keronen, P.,





Aalto, P., Hiltunen, V., Hari, P., Kulmala, M. 2009. Smart-SMEAR: on-line data exploration and visualization tool for SMEAR stations. Boreal Environment Research 14, 447–457. We thank Jan Schuladen for technical assistance with the CI-QMS and design of the TDR. We thank Thomas Klüpfel / Jonathan Williams at the Max-Planck-Institute, Mainz for the loan of a $CO_2$ detector (LICOR).

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



**Table 1: Ion source configurations according to Fig. 3 and primary-ions observed.**

| Configuration | Ion source | Source gas | Inlet gas | Primary-ions (most abundant first) |
|---|---|---|---|---|
| i | $^{210}$Po | $CH_3I + N_2$ | Syn. air | $I^-$, $I^-(H_2O)$ |
| ii | $^{210}$Po | $CH_3I + Air$ | Syn. air | $I^-$, $I^-(H_2O)$, $O_2^-$, $O_2^-(H_2O)$, $CO_3^-$ |
| iii | RF discharge | $CH_3I + N_2$ | Syn. air | $I^-$, $I^-(H_2O)$, $I(CN)_2^-$, $CNO^-$, $NO_3^-$, $IO_3^-$ |
| iv | RF discharge | $CH_3I + N_2$ | $N_2$ | $I^-$, $I^-(H_2O)$, $I(CN)_2^-$, $IH(CN)^-$ |
| v | RF discharge | $I_2 + N_2$ | Syn. air | $I^-$, $I^-(H_2O)$, $IO_3^-$, $IO_4^-$, $IO_2^-$, $NO_3^-$ |





**Table 2: Sensitivity and limit of detection of the CI-QMS for PAN, PAA, acetic acid, ClNO₂, SO₂ and HCl.**

| Reactant | Product | *m/z* | RF discharge ion source | | | ²¹⁰Po ion source | |
|---|---|---|---|---|---|---|---|
| | | | Yield (%) | $S^a$ | $LOD^b$ | $S^a$ | $LOD^b$ |
| PAN | $CH_3CO_2^-$ | 59 | 98 | 1.04 | $34^c$ | 17 | 3 |
| | $I(CN)CH_3CO_2^-$ | 212 | 2 | | | | |
| PAA | $CH_3CO_2^-$ | 59 | 98 | | | 5.7 | 4 |
| | | | | $0.22^d$ | $194^d$ | | |
| Acetic acid | $CH_3CO_2^-$ | 59 | 95 | 0.62 | 57 | n/a | n/a |
| | $CH_3C(O)OH^-$ | 60 | 3 | | | | |
| | $I(CN)CH_3CO_2^-$ | 212 | 2 | | | | |
| ClNO₂ | $ICINO_2^-$ | 208 | 30 | 0.60 | 12 | 3.5 | 3 |
| | $ICINO_2^-$ | 210 | 10 | | | | |
| | $ICl^-$ | 162 | 45 | | | | |
| | $ICl^-$ | 164 | 15 | | | | |
| SO₂ | $ISO_3^-$ | 207 | 8 | 0.09 | 56 | n/a | n/a |
| | $ISO_4^-$ | 223 | 2 | | | | |
| | $SO_3^-$ | 80 | 10 | | | | |
| | $HSO_3^-$ | 81 | 17 | | | | |
| | $SO_4^-$ | 96 | 19 | | | | |
| | $HSO_4^-$ | 97 | 30 | | | | |
| | $SO_5^-$ | 112 | 9 | | | | |
| | $HSO_5^-$ | 113 | 5 | | | | |
| HCl | $I(CN)Cl^-$ | 188 | 18 | 0.14 | 135 | n/a | n/a |
| | $I(CN)Cl^-$ | 190 | 6 | | | | |
| | $ICl^-$ | 162 | 6 | | | | |
| | $ICl^-$ | 164 | 2 | | | | |
| | $Cl^-$ | 35 | 51 | | | | |
| | $Cl^-$ | 37 | 17 | | | | |

5    [a] Sensitivity S (in Hz $ppt^{-1}$) at 50% RH (25 °C), normalised to $10^6$ Hz $I^-$ and de-clustering set to 20 V. [b] Limit of detection (LOD, 2σ, 1 s integration time). [c] The LOD is calculated from background signal at *m/z* 59 and does not include the reduction in detection limit incurred when acetic acid and PAA are present at the same *m/z* as described in the text. [d] Without de-clustering applied.



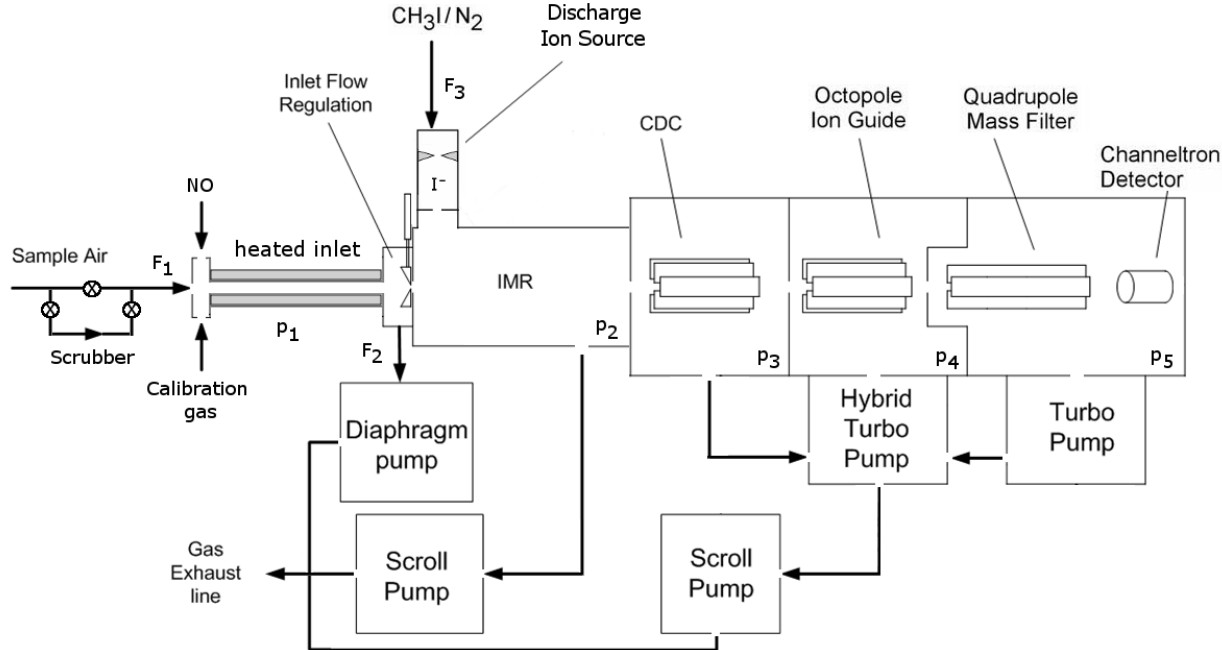

**Figure 1:** Schematic diagram of the CI-QMS. The air is sampled through the TDR (heated inlet) and enters the IMR after optional bypassing through the scrubber and mixing with calibration gas or nitrogen oxide (NO) for PAN background measurement. Ions are guided to the detector via CDC, OCT and QMF. Typical flows (F) and pressures (p) are $F_1 = 2.2$ slm, $F_2 = 1.0$ slm, $F_3 = 0.8$ slm, $p_1 =$ ambient pressure, $p_2 = 24$ mbar, $p_3 = 0.6$ mbar, $p_4 = 6 \times 10^{-3}$ mbar, $p_5 = 9 \times 10^{-5}$ mbar.





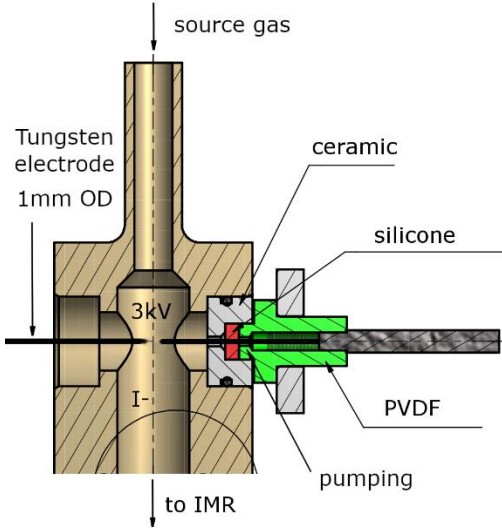

**Figure 2:** Schematic drawing of the RF-discharge ion source. The high voltage and the distance between the tips of the tungsten electrodes are variable, as described in the text. PVDF is polyvinylidene fluoride. The region around the electrodes can be optionally pumped as described in the text.

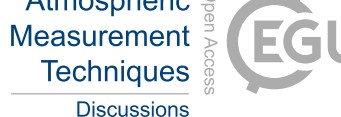

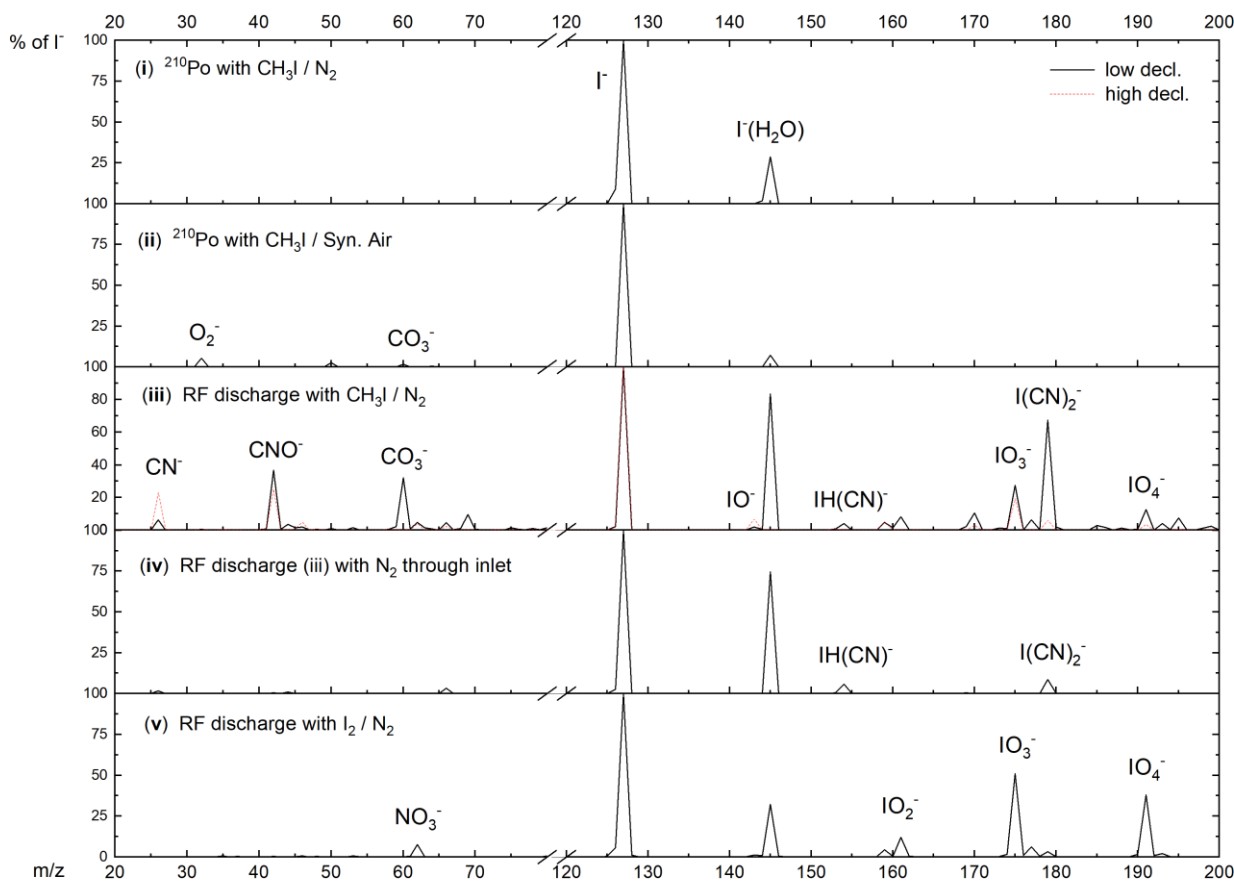

**Figure 3:** Primary-ion spectra (using $I_2$ / $N_2$ (v) or $CH_3I$ / $N_2$ (iii–iv) and using either pure synthetic air (iii) or $N_2$ (iv) for the inlet gas) obtained using the RF discharge (iii–v) or $^{210}Po$ (i–ii) as ion source. Shown are the integrated counts (8 channels) for each *m/z* normalised to the highest peak present, which is $I^-$. For this reason, true peak shapes are not visible. A description of the different configurations can be found in Table 1.



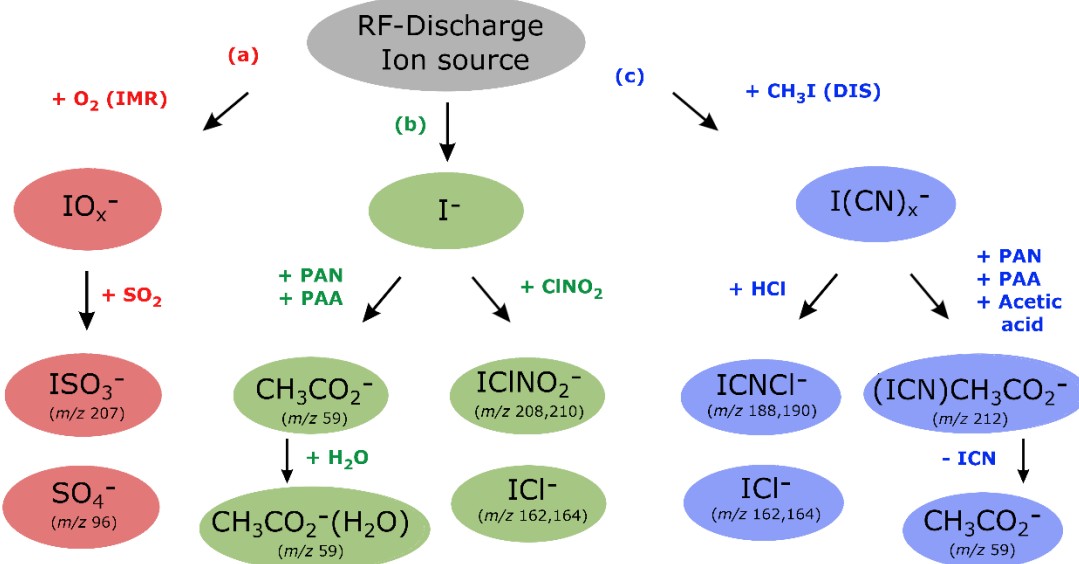

**Figure 4:** Ion detection schemes for $SO_2$, PAN, PAA, acetic acid, $ClNO_2$ and HCl using the RF discharge ion source.





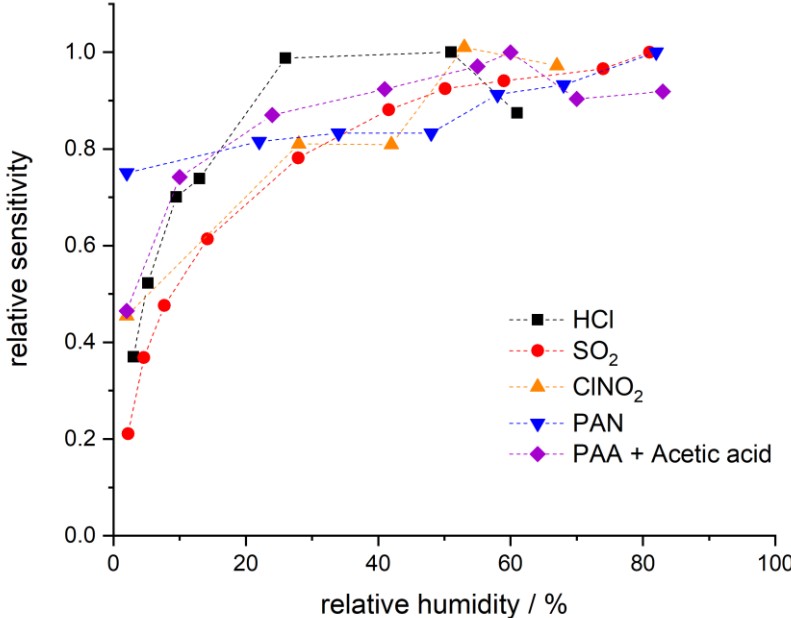

**Figure 5:** Dependence of ion signal on the relative humidity (at 25 °C and 1 bar) of the air sampled. For PAA + acetic acid on *m/z* 59 only
a combined humidity dependence is given as their contributions to the signal could not reliably be quantified.





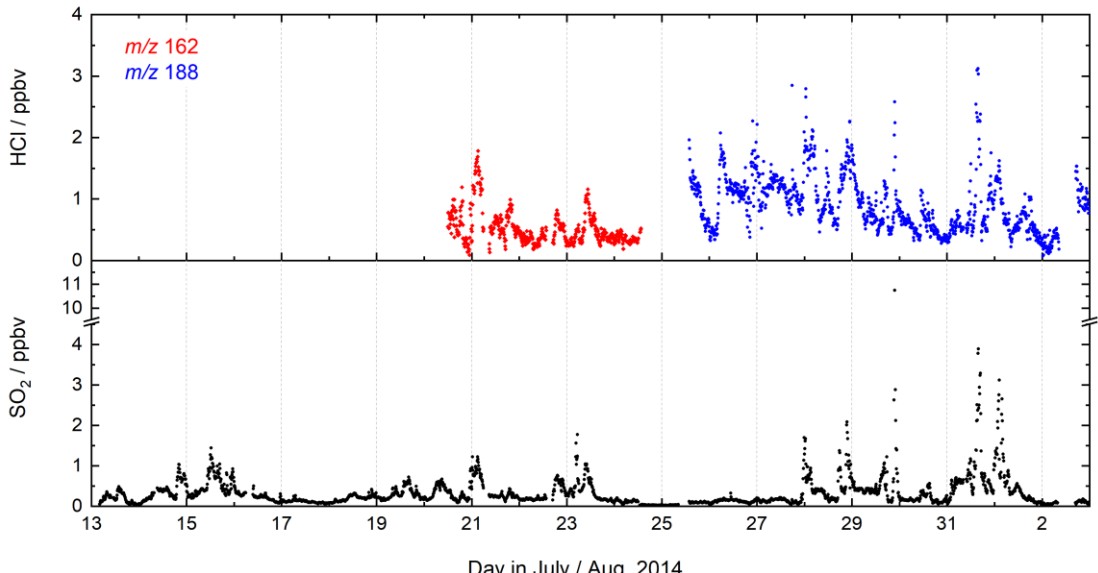

**Figure 6:** CI-QMS time series of $SO_2$ and HCl mixing ratios during the CYPHEX field campaign in Cyprus.

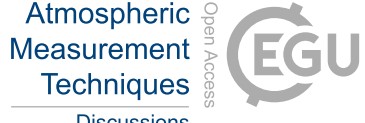



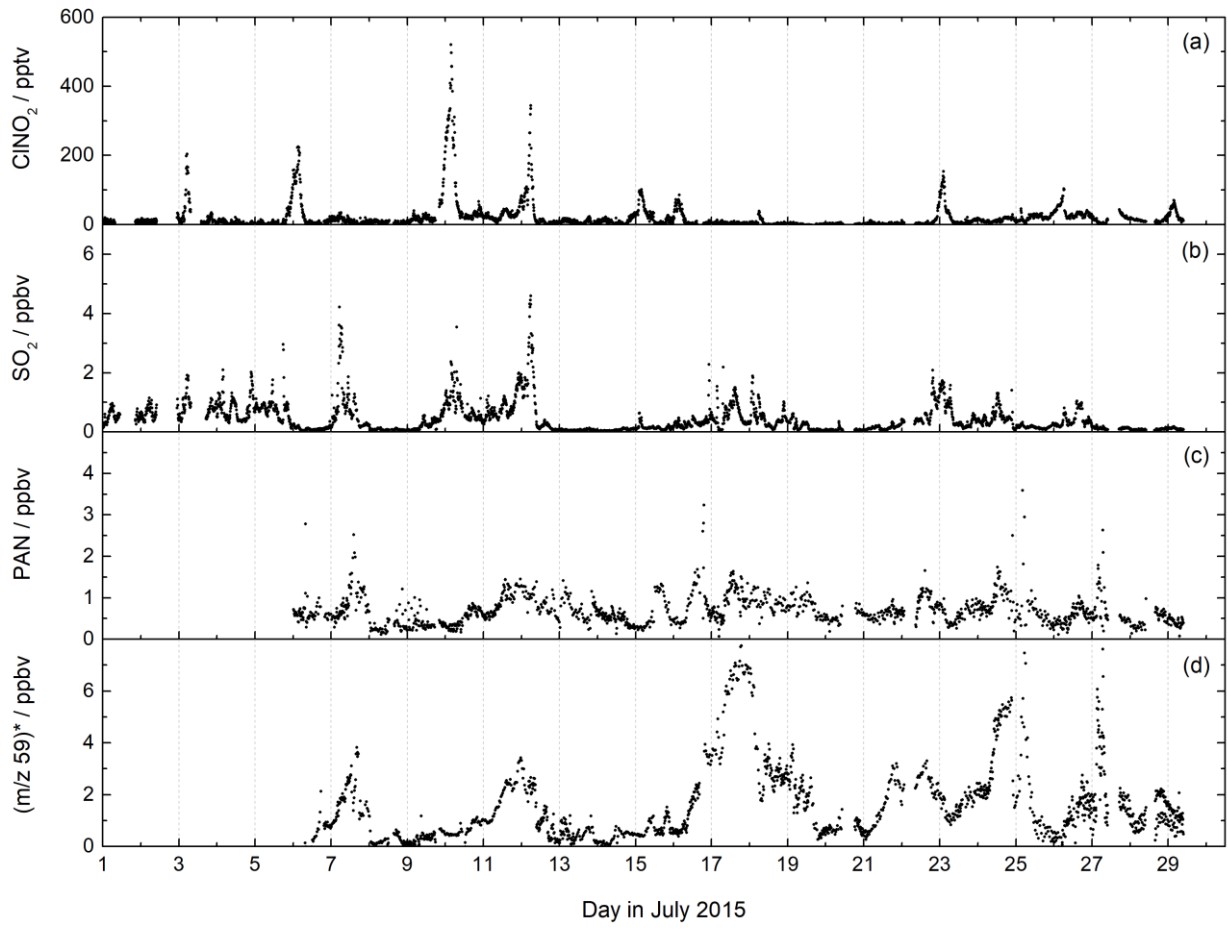

**Figure 7:** CI-QMS time series of ClNO₂, SO₂, PAN mixing ratios and *m/z* 59 during the NOTOMO field campaign in Germany. The signal at *m/z* 59 was converted to ppbv assuming that it consists only of acetic acid (no peracetic acid). The mixing ratios are therefore only an upper limit. For comparison with PARADE campaign at the same location see Fig. S6 in the supplement.





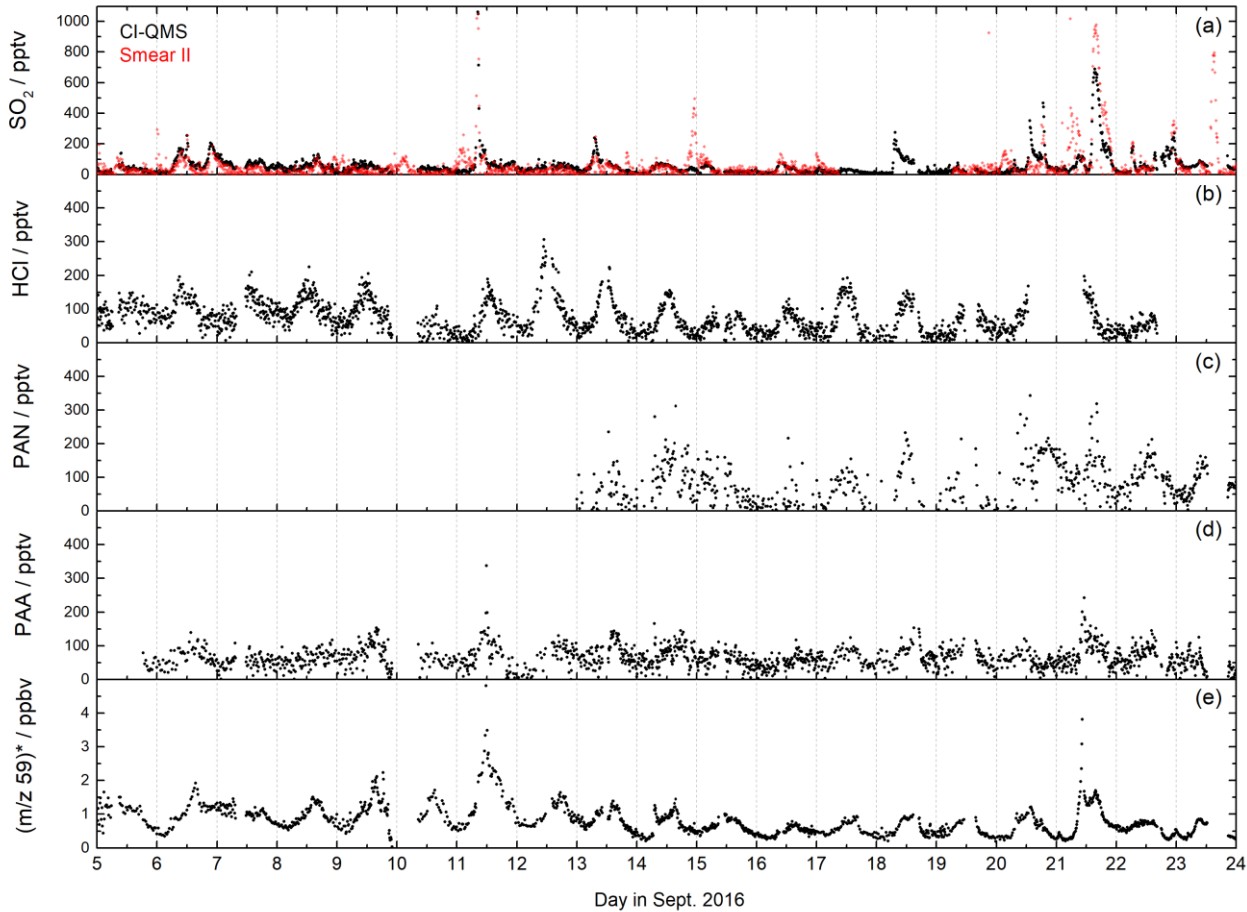

**Figure 8:** CI-QMS time series of $SO_2$, HCl, PAN, PAA mixing ratios and *m/z* 59 during the IBAIRN campaign in the boreal forest. The red $SO_2$ trace was obtained using a TEI 43 CTL fluorescence analyser (Smear II). The signal at *m/z* 59 was converted to ppbv assuming that it consists only of acetic acid (no peracetic acid). The mixing ratios are therefore only an upper limit.