# Peer review of "Chemical ionisation quadrupole mass spectrometer with an electrical discharge ion source for atmospheric trace gas measurement"

_Atmospheric Measurement Techniques, 2018_

## Referee Comment (RC1)

Review of AMT-2018-428

This paper presents an alternative ionization source for the Iodide ion TD-CIMS that does not involve radioactive material, and provides sensitivity for additional species, acetic acid, HCl and SO2. This is a useful extension of this method, and should be publishable, contingent on the author address several issues, outlined below.

General Comments

This group seems to be the only one operating an iodide ion TD-CIMS that has sensitivity to PAA. Warneke et al, (2016) describe an absence of any additional signal when NO is added to titrate PA radicals in such a TD-CIMS. It would be good to acknowledge this difference and to hear if the authors have any ideas or explanations for this.

Specific Comments

Page 2, Line 17. This would be a good place to discuss the sensitivity to PAA and possible reasons for the differences from other TD-CIMS instruments.

Page 4, Line 14 and Figure 2. It would be good to have some additional details here about the discharge ion source. What is the body of the source made of (shown in light brown in Figure 2)? Also, it is not clear, because of the material, but it is implied that the left hand side electrode is at ground, is that correct?

Page 7, Line 27. Is it true that all the combinations of sources have maximum count rates of $10^6$-$10^7$ Hz?

Page 10, Lines 12-23. This section is a repeat of previous material.

Page 11, Lines 1-2. Are you referring to the signal with NO added to titrate PA radicals?

Page 11, Line 3. A factor of 2.5 higher than what?

Page 11, Lines 7-8. Somewhere along here it seems essential that the authors present a timeline for m/z 59 that shows what it looks like when the instrument is switched between modes, e.g. TD, TD + NO added, TD+NO added +both de-clustering modes. That way we can get a sense of how the system really operates.

Page 12, lines 2-5. Do you have a more complete description of this calibration method? How was the oxidation done?

Page 14, Line 13. Don't Lee et al. address the detection of $HNO_3$ as the iodide cluster?

Page 15, Line 14. In practice, don't you use the signal through the zero catalyst as the instrument background? Is this the background for each mode?

Page 19. Lines 22-23. Doesn't $NH_4Cl$ dissociation happen in the TD inlet? This would seem to be a major interference in the measurement of HCl.

Page 19, Lines 26-27. This combination of PAA and acetic acid signals is only true if NO is being added to titrate PA radicals. This is where it would help to have one timeline figure that has the various modes labeled for the different conditions.

Figure 4. The m/z of the cluster $CH_3CO_2$-water cluster should be 77.

Supplemental Material
Figure S6. Why are the y-axes of the PAN and PAA plots so compressed, they could be expanded to maxima of 1.5 ppbv and 0.3 ppbv, respectively.

References
Warneke, C., et al., Instrumentation and Measurement Strategy for the NOAA SENEX Aircraft Campaign as part of the Southeast Atmosphere Study 2013, *Atmos. Meas. Technol.,* 9, 3063-3093, doi:10.5194/amt-9-3063-2016, 2016.

---

## Referee Comment (RC2) · Anonymous Referee #1 · 11 Feb 2019

This is a well written article. Most of my comments were already addressed during the pre-review stage. I still think that the manuscript would benefit from a more detailed comparison of this new electrical discharge method with the existing corona ion sources, but it's not absolutely necessary. In my opinion, the paper is publishable once all comments by the 2 reviewers have been addressed.

Minor comments pg 7 - "Mathieu differential equations" please provide a reference Figure 8 - have you plotted the CIMS against the SMEAR $SO_2$ data? Consider adding the fit parameters (slope, intercept, r) to the text.

A comment on the other reviewer's comment regarding the MPI group being the only

one to report PAA data: The detection of PAA by iodide CIMS is chemically similar to detection of peroxynitric acid (PNA) that was reported by Veres et al. ACP 2015, 15(4), 8101. In fact, my group's iodide CIMS is very sensitive to PAA - we just haven't reported any data as we havn't developed a calibration source to determine response factors and customized the inlet setup as the Crowley group has done.
* * *

---

## Author Comment (AC1) · 19 Feb 2019

The comment was uploaded in the form of a supplement:
https://www.atmos-meas-tech-discuss.net/amt-2018-428/amt-2018-428-AC1-supplement.pdf

---

## Author Response (AR1)

**Response to review of referee #1**

*In the following, the referee's comments are reproduced (black) along with our replies (blue)*
*and changes made to the text (red) in the revised manuscript.*

**General statement:**

This is a well written article. Most of my comments were already addressed during the pre-review stage. I still think that the manuscript would benefit from a more detailed comparison of this new electrical discharge method with the existing corona ion sources, but it's not absolutely necessary. In my opinion, the paper is publishable once all comments by the 2 reviewers have been addressed.

We thank the referee for the positive evaluation of our manuscript which we modified according to the comments listed below.

A detailed comparison of our ion-source with existing corona ion sources operating with $I^-$ primary ions would indeed be most useful. However, we are unaware of other CIMS instruments using iodide with a corona source. We now write:
Potential alternatives are corona discharge and x-ray ion sources as commonly used in atmospheric pressure chemical ionisation mass spectrometers (AP-CIMS) (Jost et al., 2003; Skalny et al., 2007; Kürten et al., 2011; Zheng et al., 2015). We are unaware of previous usage of corona discharges for the generation of iodide ions.

**General comments:**

A comment on the other reviewer's comment regarding the MPI group being the only one to report PAA data: The detection of PAA by iodide CIMS is chemically similar to detection of peroxynitric acid (PNA) that was reported by Veres et al. ACP 2015, 15(4), 8101. In fact, my group's iodide CIMS is very sensitive to PAA - we just haven't reported any data as we havn't developed a calibration source to determine response factors and customized the inlet setup as the Crowley group has done.

We are not the only group to detect PAA using $I^-$ primary ions.
The discussion to our original ACP paper on PAN and PAA detection (Phillips et al, Atmos. Chem. Phys., 13, 1129-1139, doi:10.5194/acp-13-1129-2013, 2013.) confirmed sensitivity of other Iodide-CIMS to ambient PAA.
In addition, Furgeson et al. (2011) report detection of PAA. The following text has been added:

Unlike other TD-CIMS instruments that describe an absence of a residual signal when NO is added to the inlet (e.g. Warnecke et al., 2016), the I-CIMS deployed by Phillips et al. (2013) as well as the instrument presented in this study are very sensitive to PAA at m/z 59. Furgeson et al. (2011) also describe an interference at m/z 59 that is not titrated by NO and suggest detection of PAA which is produced in their photochemical source used for PAN generation. In addition, Veres et al. (2015) report a very similar mechanism to PAA detection for the detection of pernitric acid (PNA). Differences in the sensitivities of various I-CIMS instruments to PAA at m/z 59 are likely to be associated with different de-clustering potentials.

**Specific comments:**

pg 7 - "Mathieu differential equations" please provide a reference
We deleted the sentence regarding "Mathieu differential equations" as it is not of relevance for our manuscript.

Figure 8 - have you plotted the CIMS against the SMEAR SO2 data? Consider adding the fit parameters (slope, intercept, r) to the text.
This is not warranted. The point of this figure was not to make a detailed inter-comparison of the two $SO_2$ instruments, which were not co-located with the SMEAR instrument mounted above the canopy. In addition, much of the data reported by the SMEAR instrument are below its detection limit (0.1 ppbv) and only occasional plumes of $SO_2$ are observed.

*In the following, the referee's comments are reproduced (black) along with our replies (blue)*
*and changes made to the text (red) in the revised manuscript.*

**General statement:**

This paper presents an alternative ionization source for the Iodide ion TD-CIMS that does not involve radioactive material, and provides sensitivity for additional species, acetic acid, HCl and SO2. This is a useful extension of this method, and should be publishable, contingent on the author address several issues, outlined below.

We thank the referee for the positive evaluation of our manuscript which we modified according to the comments listed below.

**General comments:**

This group seems to be the only one operating an iodide ion TD-CIMS that has sensitivity to PAA. Warneke et al, (2016) describe an absence of any additional signal when NO is added to titrate PA radicals in such a TD-CIMS. It would be good to acknowledge this difference and to hear if the authors have any ideas or explanations for this.

Page 2, Line 17. This would be a good place to discuss the sensitivity to PAA and possible reasons for the differences from other TD-CIMS instruments.

We are not the only group to detect PAA using I⁻ primary ions.
The discussion to our original ACP paper on PAN and PAA detection (Phillips et al, Atmos. Chem. Phys., 13, 1129-1139, doi:10.5194/acp-13-1129-2013, 2013.) confirmed sensitivity of other Iodide-CIMS to ambient PAA. In addition, Furgeson et al. (2011) report detection of PAA.

The following text has been added:

Unlike other TD-CIMS instruments that describe an absence of a residual signal when NO is added to the inlet (e.g. Warnecke et al., 2016), the I-CIMS deployed by Phillips et al. (2013) as well as the instrument presented in this study are very sensitive to PAA at m/z 59. Furgeson et al. (2011) also describe an interference at m/z 59 that is not titrated by NO and suggest detection of PAA which is produced in their photochemical source used for PAN generation. In addition, Veres et al. (2015) report a very similar mechanism to PAA detection for the detection of pernitric acid (PNA). Differences in the sensitivities of various I-CIMS instruments to PAA at m/z 59 are likely to be associated with different de-clustering potentials.

**Specific comments:**

Page 4, Line 14 and Figure 2. It would be good to have some additional details here about the discharge ion source. What is the body of the source made of (shown in light brown in Figure 2)? Also, it is not clear, because of the material, but it is implied that the left hand side electrode is at ground, is that correct?

We now provide additional information in the caption of Fig. 2. Furthermore, the figure has been updated to make it clear that both electrodes are connected to a transformer.

A transformer, which is supplied with up to 200 V AC from the internal V25 unit, applies temporarily fluctuating potentials to both electrodes. The body of the ion source (coloured in light brown) is made of stainless steel.

Page 7, Line 27. Is it true that all the combinations of sources have maximum count rates of $10^6$-$10^7$ Hz?

Yes, that is true. With all configurations ($Po^{210}$ ioniser, discharge ion source (DIS) + $CH_3I$ and DIS + $I_2$) maximum count rates in the range of $(6 \pm 2)$ x$10^6$ Hz were achieved.

The text now states:

The absolute ion count rates for both ion sources are comparable, i.e. up to $(6 \pm 2)$ x $10^6$ Hz for I-.

Page 10, Lines 12-23. This section is a repeat of previous material.

We have rearranged the material provided in Sect. 2.1 and 4.1 to avoid duplication and to improve the structure of the manuscript.

"The rate coefficient for the thermal decomposition of PAN (at 453 K and 1 bar) is □ 2000 s-1 (Atkinson et al., 2006; IUPAC, 2018), so that > 99.99 % of PAN should be thermally dissociated within 200 ms. This could be confirmed by measurement of the signal due to a stable PAN source whilst varying the inlet temperature."

…has been moved from Sect 2.1 to Sect. 4.1.

"In order to discriminate between PAN and PAA (both measured at m/z 59) in ambient air a known concentration of NO is added at the front end of the TDR to remove the CH3C(O)O2 radicals formed after thermal decomposition of PAN (see Sect. 4.1)."

…has been deleted.

Page 11, Lines 1-2. Are you referring to the signal with NO added to titrate PA radicals?
Yes, NO has to be added to remove the PAN signal so that only PAA + AA remain. We inserted text to clarify this:
When NO is added, […]

Page 11, Line 3. A factor of 2.5 higher than what?
This refers to higher than with de-clustering. We now write:
Unfortunately, the chemical background without de-clustering at *m/z* 59 is by a factor of 2.5 higher than with de-clustering […]

Page 11, Lines 7-8. Somewhere along here it seems essential that the authors present a timeline for m/z 59 that shows what it looks like when the instrument is switched between modes, e.g. TD, TD + NO added, TD+NO added +both de-clustering modes. That way we can get a sense of how the system really operates.
Good idea. We have added Figure S3 to the manuscript which we refer to in the text:
An exemplary time series showing the different measurement modes of the CI-QMS (Scrubber, Ambient, NO addition) and the differences in detection of PAA and acetic acid when applying a de-clustering voltage is provided in Fig. S3 of the supplementary information.

Page 12, lines 2-5. Do you have a more complete description of this calibration method? How was the oxidation done?
We have inserted a reference describing the method in detail:
Veres, P., Gilman, J. B., Roberts, J. M., Kuster, W. C., Warneke, C., Burling, I. R., & Gouw, J. D. (2010). Development and validation of a portable gas phase standard generation and calibration system for volatile organic compounds. Atmospheric Measurement Techniques, 3(3), 683-691.

Page 14, Line 13. Don't Lee et al. address the detection of $HNO_3$ as the iodide cluster?
Correct. We have inserted a reference to Lee et al. (2014):
Lee, B. H., Lopez-Hilfiker, F. D., Mohr, C., Kurtén, T., Worsnop,D. R., and Thornton, J. A.: An Iodide-Adduct High-Resolution Time-of-Flight Chemical-Ionization Mass Spectrometer: Application to Atmospheric Inorganic and Organic Compounds, Environ. Sci. Technol., 48, 6309–6317, doi:10.1021/es500362a, 2014.

Page 15, Line 14. In practice, don't you use the signal through the zero catalyst as the instrument background? Is this the background for each mode?

Yes, in practise we use the zero catalyst (Scrubber) for BG determination. This BG is the same for each mode as the addition of NO does not change the zero signals.

The limit of detection (LOD) is mainly dependent on variability in the background signal on the respective *m/z* and can be calculated as two times the standard deviation when using synthetic (i.e. hydrocarbon-free) air or when passing the air through the scrubber (as usually performed during field measurements).

Page 19. Lines 22-23. Doesn't $NH_4Cl$ dissociation happen in the TD inlet? This would seem to be a major interference in the measurement of HCl.

In all our ambient air measurements we used a particle filter in front of the sampling line. We have inserted a line at the beginning of Sect. 2.1 ("TDR") to clarify this:

Ambient air entering the TDR first passes through a 2 μm pore size membrane filter (Pall Teflo) to efficiently remove aerosols.

Page 19, Lines 26-27. This combination of PAA and acetic acid signals is only true if NO is being added to titrate PA radicals. This is where it would help to have one timeline figure that has the various modes labeled for the different conditions.

We inserted:

"When NO is added (see Fig. S3), […] "

and added Fig. S3 (see above)

Figure 4. The m/z of the cluster $CH_3CO_2$-water cluster should be 77.

The labelling in Figure 4 was corrected.

Supplemental Material

Figure S6. Why are the y-axes of the PAN and PAA plots so compressed, they could be expanded to maxima of 1.5 ppbv and 0.3 ppbv, respectively.

The scaling of the y-axes in Fig. S6 was modified.

References

Warneke, C., et al., Instrumentation and Measurement Strategy for the NOAA SENEX Aircraft Campaign as part of the Southeast Atmosphere Study 2013, *Atmos. Meas. Technol.,* 9, 3063-3093, doi:10.5194/amt-9-3063-2016, 2016.

The reference was added to the manuscript, see above.

Revised manuscript (changes in yellow):

[revised manuscript text omitted]

**Supplement**

[Figure]

[Figure]

**Figure S1**. Photo: $N_2$ emission observed between and around the pointed tungsten tips of the electrodes of the RF discharge source. Right: The emission spectrum was recorded with an Ocean-Optics USB-4000 spectrometer with optical fibre at various high-voltages. The strongest features (not fully resolved using the low-resolution ($\Delta\lambda \approx 1.5$ nm) spectrograph) can be assigned to transitions from the ground vibrational level of the electronically excited $N_2$ ($C^3\Pi_u$) state to the $B^3\Pi_g$ state.

[Figure]

**Figure S2**. (a): Dependence of $IO_3^-$ signal ($m/z$ 175) on the fractional pressure of $O_2$ in the IMR when adding 800 sccm $N_2$ / $CH_3I$ through the RF discharge region. (b): Signal at $m/z$ 207 ($ISO_3^-$) for a constant amount of $SO_2$ over the same range of $O_2$ partial pressures.

[Figure]

**Figure S3**. Exemplary time series of the CI-QMS signal at *m/z* 59 when sampling from the photochemical PAN calibration source described in Sect. 2.7. The instrument periodically switches between the states "scrubber", "ambient" and "titration" (i.e. addition of NO). In ambient mode we measure the total signal of PAN, PAA and acetic acid. When adding NO to the inlet, the peroxyacyl radicals are titrated and the signal consists of PAA and acetic acid only. Changing the de-clustering voltage (at 15:00) from 20 V to 2 V results in the complete loss of sensitivity for acetic acid. During measurements of ambient air the de-clustering voltage is usually changed more frequently, which means for each data point we firstly measure the signal at *m/z* 59 with high de-clustering and immediately afterwards with low de-clustering.

[Figure]

**Figure S4**. (a) Linear dependence of count rate at $m/z$ 207 (ISO$_3^-$) on the SO$_2$ mixing ratio of the sample measured. (b) Linear dependence of count rate at $m/z$ 188 (I(CN)Cl$^-$) on the HCl mixing ratio.

[Figure]

**Figure S5**. (a) and (b): Correlation of ion signals at $m/z$ 162 versus $m/z$ 164 (ICl$^-$) and $m/z$ 188 versus $m/z$ 190 (I(CN)Cl$^-$) during CYPHEX. The expected slope resulting from the isotopic abundance of $^{35}$Cl to $^{37}$Cl is 3.13. (c) Signal at $m/z$ 188 versus $m/z$ 162. The linear correlation indicates that both ions are from the same trace gas, HCl.

[Figure]

5   **Figure S6:** Correlation between the CI-QMS measurement of $SO_2$ at $m/z$ 207 ($ISO_3^-$) vs. $m/z$ 97 ($HSO_4^-$) during NOTOMO.

[Figure]

**Figure S7:** Measurements of $ClNO_2$, PAN and PAA using CI-QMS with a [210]Po-ionisation source during the PARADE campaign, which took place at the same location and similar time of year as the NOTOMO campaign in which the RF discharge was deployed. The $ClNO_2$ data during PARADE has been reported by Phillips et al. (2012).